# DAWN: Dynamic Frame Avatar with Non-autoregressive Diffusion Framework for talking head Video Generation

**Hanbo Cheng**[1][*]**, Limin Lin**[1][*]**, Chenyu Liu**[2]**, Pengcheng Xia**[2]**, Pengfei Hu**[1]**,**
**Jiefeng Ma**[1]**, Jun Du**[1][†]**, Jia Pan**[2]

[1]University of Science and Technology of China, [2]iFLYTEK Research

https://hanbo-cheng.github.io/DAWN

## ABSTRACT

Talking head generation intends to produce vivid and realistic talking head videos from a single portrait and speech audio clip. Although significant progress has been made in diffusion-based talking head generation, almost all methods rely on autoregressive strategies, which suffer from limited context utilization beyond the current generation step, error accumulation, and slower generation speed. To address these challenges, we present DAWN (**D**ynamic frame **A**vatar **W**ith **N**on-autoregressive diffusion), a framework that enables all-at-once generation of dynamic-length video sequences. Specifically, it consists of two main components: (1) audio-driven holistic facial dynamics generation in the latent motion space, and (2) audio-driven head pose and blink generation. Extensive experiments demonstrate that our method generates authentic and vivid videos with precise lip motions, and natural pose/blink movements. Additionally, with a high generation speed, DAWN possesses strong extrapolation capabilities, ensuring the stable production of high-quality long videos. These results highlight the considerable promise and potential impact of DAWN in the field of talking head video generation. Furthermore, we hope that DAWN sparks further exploration of non-autoregressive approaches in diffusion models. Our code will be publicly available at `https://github.com/Hanbo-Cheng/DAWN-pytorch`.

## 1 INTRODUCTION

Talking head generation aims at synthesizing a realistic and expressive talking head from a given portrait and audio clip, which is garnering growing interest due to its potential applications in virtual meetings, gaming, and film production. For talking head generation, it is essential that the lip motions in the generated video precisely match the accompanying speech, while maintaining high overall visual fidelity (Guo et al., 2021a). Furthermore, natural coordination between head pose, eye blinking, and the rhythm of the audio is also crucial for a convincing output (Liu et al., 2023).

Recently, Diffusion Models (DM) (Ho et al., 2020) have achieved significant success in video and image generation tasks (Rombach et al., 2022; Ho et al., 2022b;a; Peebles & Xie, 2023; Ni et al., 2023). However, their application in talking head generation (Shen et al., 2023; Bigioi et al., 2024) still faces several challenges. Although many methods yield high-quality results, most of them rely on autoregressive (AR) (Tian et al., 2024; Ma et al., 2023) or semi-autoregressive (SAR) (Xu et al., 2024c; He et al., 2023) strategies. In each iteration, AR generates one frame, while SAR generates a fixed-length video segment. The two strategies significantly slow down the inference speed and fail to adequately utilize contextual information from future frames, which leads to constrained performance and potential error accumulation, especially in long video sequences (Stypułkowski et al., 2024; Tian et al., 2024; Xu et al., 2024b).

---

[*]Authors contributed equally to this research, work done as interns at iFLYTEK Research.
[†]Corresponding author: Jun Du (jundu@ustc.edu.cn).

To address these challenges, we present DAWN (**D**ynamic frame **A**vatar **W**ith **N**on-autoregressive diffusion), a novel approach that significantly improves both the quality and efficiency of talking head generation. Our approach leverages the DM to generate motion representation sequences from a given audio and portrait. These motion representations are subsequently used to reconstruct the video. Unlike other methods, our approach produces videos of arbitrary length in a non-autoregressive (NAR) manner. However, employing the NAR strategy to generate long videos often results in either over-smoothing or significant content inconsistencies due to limited extrapolation (Qiu et al., 2023). In the context of talking head generation, we suggest that the model's temporal modeling capability is significantly hindered by the strong coupling relationship among multiple motions. Typically, the motions in talking head include (1) lip motions and (2) head pose and blink movements. The temporal dependency of head and blink movements extends over several seconds, far longer than that of lip motions (Zhou et al., 2020). Training models to capture these long-term dependencies requires extensive sequences, thus increasing the difficulty and cost of training. Fortunately, head and blink movements can be represented as low-dimensional vectors (Zhang et al., 2023), enabling the design of a lightweight model that learns these long-term dependencies by training on extended sequences. Thus, to further enhance the temporal modeling and extrapolation capabilities of DAWN, we disentangle the motion components involved in talking head videos. Specifically, we use the Audio-to-Video Flow Diffusion Model (A2V-FDM) to learn the implicit mapping between the lips and audio, while generating the head pose and blinks via explicit control signals. Additionally, we propose a lightweight Pose and Blink generation Network (PBNet) trained on long sequences, to generate natural pose/blink movements during inference in a NAR manner. In this way, we simplify the training of A2V-FDM as well as achieve the long-term dependency modeling of the pose/blink movement. To further strengthen the convergence and extrapolation capabilities of A2V-FDM, we propose a two-stage training strategy based on curriculum learning to guide the model in generating accurate lip motion and precise pose/blink movement control.

The main contributions of this work are as follows: 1) We present DAWN (**D**ynamic frame **A**vatar **W**ith **N**on-autoregressive diffusion) for generating dynamic-length talking head videos from portrait images and audio clips in a non-autoregressive (NAR) manner, achieving faster inference speeds and high-quality results. To the best of our knowledge, this is the first NAR solution based on diffusion models designed for general talking head generation. 2) To compensate for the limitations of extrapolation in NAR strategies and enhance the temporal modeling capabilities for long videos, we decouple the motions of the lips, head, and blink, achieving precise control over these movements. 3) We propose the Pose and Blink generation Network (PBNet) to generate natural head pose and blink sequences exclusively from audio in a NAR manner. 4) We introduce the Two-stage Curriculum Learning (TCL) strategy to guide the model in mastering lip motion generation and precise pose/blink control, ensuring strong convergence and extrapolation ability.

## 2 RELATED WORKS

**Audio-driven talking head generation.** Initial approaches for talking head generation employed deterministic models to map audio to video streams (Fan et al., 2016), with later methods introducing generative models such as GANs (Isola et al., 2016a), VAEs (Kingma & Welling, 2022), and diffusion models (DMs) (Ho et al., 2020). GAN-related methods (Vougioukas et al., 2020; Pumarola et al., 2018; Hong et al., 2022) improved visual realism but faced convergence and mode collapse issues (Xia et al., 2022). Following this, VAEs (Kingma & Welling, 2022) generated 3D priors like 3D Morphable Models (3DMM) (Blanz & Vetter, 1999) followed by high-fidelity rendering (Ren et al., 2021), which limited the realism and vividness. In contrast, DMs have been introduced into talking head generation due to their good convergence, excellent generation performance, and diversity. Stypułkowski et al. (2024) presented a DM-based talking head generation solution using an AR strategy to generate videos frame-by-frame iteratively. Subsequently, Tian et al. (2024) improved this AR strategy by incorporating motion conditions extracted by a VAE-based network as priors for each iteration, effectively mitigating degradation issues. Concurrently, Xu et al. (2024c); He et al. (2023) advocated for motion modeling instead of image modeling. They utilized DMs to iteratively generate latent motion representations over a fixed number of frames in a SAR manner, subsequently converting these motion representations into video frames. While most diffusion-based methods produce promising results, their AR or SAR strategies incur slow generation speeds and collapse in long-video generation. Although methods like Tian et al. (2024); Xu et al. (2024c) alleviate issues such as inconsistencies in content across iterations and long video generation collapse, the risk of

error accumulation remains unresolved. While methods like Du et al. (2023) use identity-specific NAR strategies, to the best of our knowledge, none have addressed NAR talking head generation for arbitrary identities. Consequently, we propose a novel NAR dynamic frame generation framework based on DM, which aims to achieve the low-cost and high-quality rapid generation of realistic talking head video through a clip of audio and arbitrary portrait.

**Audio-driven pose and blink generation.** Head pose and blink movements significantly impact the naturalness of talking head videos. However, the mapping from audio to pose and blink movement is a one-to-many problem, which presents a significant challenge (Xu et al., 2024a; Chen et al., 2020). Early works primarily focused on controlling poses directly using facial landmarks or video references (Zhou et al., 2021; Guo et al., 2021b). However, these approaches require additional guidance information, which impairs the diversity of the results. Later studies considered generating both pose and blink within the context of talking head generation (Zhou et al., 2020). However, simultaneously generating all facial movements can cause interference and ambiguity (Zhang et al., 2023). Therefore, some works attempted to decouple the speaker's actions into components like lip, head pose, and blink, using discriminative models to predict these conditions separately (Wang et al., 2021; He et al., 2023). Later, researchers recognized that probabilistic modeling is better suited for the one-to-many mapping relationship, leading to the proposal of a VAE-based pose generation framework (Liu et al., 2023). However, most existing pose generation strategies also depend on AR or SAR approaches, negatively impacting efficiency, smoothness, and naturalness. To address these issues, we design a VAE-based NAR pose generation method to produce vivid and smooth pose and blink movements while maintaining the NAR generation of the entire framework.

## 3 METHOD

As shown in Figure 1, DAWN is divided into three main parts: (1) the Latent Flow Generator (LFG); (2) the conditional Audio-to-Video Flow Diffusion Model (A2V-FDM); and (3) the Pose and Blink generation Network (PBNet). First, we train the LFG to estimate the motion representation between different video frames in the latent space. Subsequently, the A2V-FDM is trained to generate temporally coherent motion representation from audio. Finally, PBNet is used to generate poses and blinks from audio to control the content in the A2V-FDM. To enhance the model's extrapolation ability while ensuring better convergence, we propose a novel Two-stage Curriculum Learning (TCL) training strategy. We will first discuss preliminaries, then present the specific details of DAWN's three main components, namely LFG, A2V-FDM, and PBNet in Sections 3.2, 3.3, and 3.4, respectively. Finally, we will introduce the TCL strategy in Section 3.5.

### 3.1 PRELIMINARIES

**Task definition.** The task of talking head generation involves creating a natural and vivid talking head video from two inputs: a static single-person portrait, $\boldsymbol{x}_{\mathrm{src}}$, and a speech sequence, $\boldsymbol{y}_{1:N} = \{\boldsymbol{y}_0, \boldsymbol{y}_1, \ldots, \boldsymbol{y}_N\}$. The static image $\boldsymbol{x}_{\mathrm{src}}$ and the speech sequence $\boldsymbol{y}$ can originate from any individual, and the output is $\hat{\boldsymbol{x}}_{1:N} = \{\hat{\boldsymbol{x}}_0, \hat{\boldsymbol{x}}_1, \ldots, \hat{\boldsymbol{x}}_N\}$, where $N$ represents the total number of frames.

**Diffusion models.** Diffusion models generate samples conforming to a given data distribution by progressively denoising Gaussian noise (Ho et al., 2020). Let $\boldsymbol{x}_0$ represent data sampled from a given distribution $q(\boldsymbol{x}_0)$. In the forward diffusion process, Gaussian noise is progressively added to $\boldsymbol{x}_0$ after $T$ steps, resulting in noisy data $\boldsymbol{x}_T$ (Nichol & Dhariwal, 2021; Song et al., 2020), and the conditional transition distribution at each step is defined as:

$$q(\boldsymbol{x}_t|\boldsymbol{x}_0) = \mathcal{N}(\boldsymbol{x}_t; \sqrt{\bar{\alpha}_t}\,\boldsymbol{x}_0, (1 - \bar{\alpha}_t)\boldsymbol{I}) \tag{1}$$

where $\bar{\alpha}_t = \prod_{i=1}^{t} \alpha_i$. The reverse diffusion process gradually recovers the original data from the Gaussian noise $\boldsymbol{x}_T \sim \mathcal{N}(0, I)$, utilizing a neural network to predict $p_\theta(\boldsymbol{x}_{t-1}|\boldsymbol{x}_t)$, where $\theta$ represents the parameters of the neural network. The model is trained using the following loss function:

$$L_{\mathrm{simple}} = \mathbb{E}_{t,\boldsymbol{x}_0,\epsilon}[\|\epsilon - \epsilon_\theta(\boldsymbol{x}_t, t)\|^2] \tag{2}$$

where $\epsilon$ is the Gaussian noise added to $\boldsymbol{x}_0$ in the forward diffusion process to obtain $\boldsymbol{x}_t$, and $\epsilon_\theta(\boldsymbol{x}_t, t)$ is the noise predicted by the model. In video-related tasks, the denoising model can be implemented via a 3D U-Net (Ho et al., 2022b; Çiçek et al., 2016).

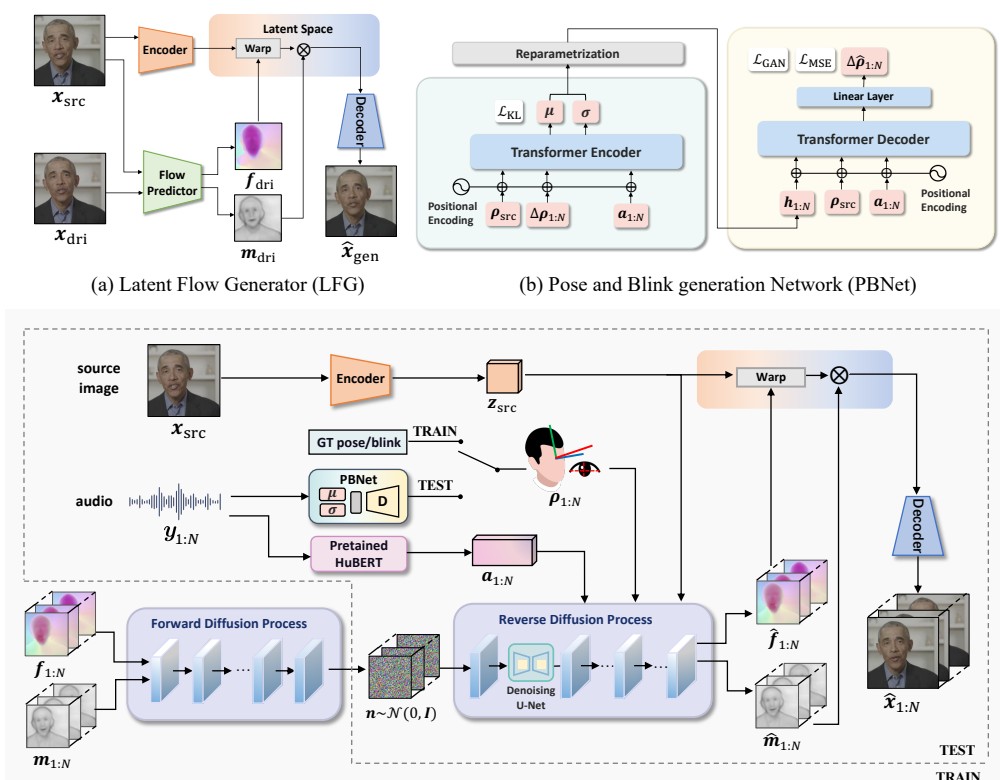

(a) Latent Flow Generator (LFG)  (b) Pose and Blink generation Network (PBNet)

(c) The structure of the **D**ynamic frame **A**vatar **W**ith **N**on-Autoregressive Diffusion Framework (**DAWN**)

Figure 1: The pipeline of DAWN. First, we train the Latent Flow Generator (LFG) in (a) to extract the motion representation from the video. Then the Pose and Blink generation Network (PBNet) in (b) is utilized to generate the head pose and blink sequences of the avatar. Subsequently, the Audio-to-Video Flow Diffusion Model (A2V-FDM) in (c) generates the talking head video from the source image conditioned by the audio and pose/blink sequences provided by the PBNet.

## 3.2 LATENT FLOW GENERATOR

The Latent Flow Generator (LFG) is a self-supervised training framework designed to model motion information between the source image $x_{\text{src}}$ and the driving image $x_{\text{dri}}$. As illustrated in Figure 1 (a), LFG consists of three trainable modules: the image encoder $\mathcal{E}$, the flow predictor $\mathcal{P}$, and the image decoder $\mathcal{D}$. During training, $x_{\text{src}}, x_{\text{dri}} \in \mathbb{R}^{H \times W \times 3}$ are images randomly selected from the same video. The image encoder $\mathcal{E}$ encodes the source image $x_{\text{src}}$ into a latent code $z_{\text{src}} \in \mathbb{R}^{H_z \times W_z \times C_z}$. The flow predictor estimates a dense flow map $f$ and a blocking map $m$ (Siarohin et al., 2021; 2020), corresponding to $x_{\text{src}}$ and $x_{\text{dri}}$ :

$$f, m = \mathcal{P}(x_{\text{src}}, x_{\text{dri}}) \tag{3}$$

The flow map $f \in \mathbb{R}^{H_z \times W_z \times 2}$ describes the feature-level movement of $x_{\text{dri}}$ relative to $x_{\text{src}}$ in horizontal and vertical directions. The blocking map $m \in \mathbb{R}^{H_z \times W_z \times 1}$ ranging from 0 to 1, indicates the degree of area blocking in the transformation from $x_{\text{src}}$ to $x_{\text{dri}}$. The flow map $f$ is used to perform the affine transformation $\mathcal{A}$, serving as a coarse-grained warping of $z_{\text{src}}$. Subsequently, the blocking map $m$ guides the model in repairing the occlusion area, thereby serving as fine-grained repair. Finally, the image decoder $\mathcal{D}$ converts the warped latent code into the target image $\hat{x}_{\text{gen}}$, where the $\otimes$ is the element-wise product:

$$\hat{x}_{\text{gen}} = \mathcal{D}(\mathcal{A}(z_{\text{src}}, f) \otimes m) \tag{4}$$

The LFG is trained in an unsupervised manner and optimized using the following reconstruction loss:

$$L_{\text{LFG}} = L_{\text{rec}}(\hat{x}_{\text{gen}}, x_{\text{dri}}) \tag{5}$$

where $L_{\text{rec}}$ is a multi-resolution reconstruction loss derived from a pre-trained VGG-19 network, used to evaluate perceptual differences between $\hat{x}_{\text{gen}}$ and $x_{\text{dri}}$ (Johnson et al., 2016). We consider the concatenation of $m$ and $f$ as $z^m = [f, m]$ to represent the motion of $x_{\text{dri}}$ relative to $x_{\text{src}}$. In this way, we achieve two objectives: 1) finding an effective explicit motion representation $z^m$, which is identity-agnostic and well-supported by physical meaning, and 2) reconstructing $x_{\text{dri}}$ from $x_{\text{src}}$ and $z^m$ without the need for a full pixel generation.

### 3.3 Conditional Audio2Video Flow Diffusion Model

Through the LFG in Section 3.2, we can specify a $x_{\text{src}}$, and extract the identity-agnostic motion representations $z^m_{1:N}$ of the talking head video clip $x_{1:N}$ as well as the latent code $z_{\text{src}}$ extracted by $\mathcal{E}$ from $x_{\text{src}}$. Therefore, we design the A2V-FDM to generate the motion representations $\hat{z}^m_{1:N} = [\hat{f}_{1:N}, \hat{m}_{1:N}]$ of each frame relative to $x_{\text{src}}$:

$$\hat{z}^m_{1:N} = \text{DM}(\mathcal{E}(x_{\text{src}}), y_{1:N}, \rho_{1:N}) \tag{6}$$

where the DM refers the diffusion model and $\rho_{1:N}$ is the pose/blink signal. After generating $\hat{z}^m_{1:N}$, we use the decoder $\mathcal{D}$ in LFG to reconstruct the $\hat{z}^m_{1:N}$ to the target video frames, via the Equation 4. The structure of A2V-FDM is illustrated in Figure 1 (c). The A2V-FDM model includes a 3D U-Net denoising backbone (Ho et al., 2022b). The residue block in the 3D U-Net contains temporal attention and spatial attention modules, which handle frame-level and pixel-level dependencies respectively. The parameters of the temporal attention module are independent of the input length, so we believe that 3D U-Net can theoretically process video sequences of any length. However, to ease training difficulty, we train on short sequences and aim for long-sequence inference. To enhance the 3D U-Net's extrapolation ability when handling long sequences, we used Rotary Positional Encoding (RoPE) (Su et al., 2023) instead of traditional absolute position embeddings in the temporal attention module.

**Conditioning**. We incorporate the following conditions to control its generative behavior: audio embedding $a_{1:N}$, pose/blink signal $\rho_{1:N}$, and source image latent code $z_{\text{src}}$. The audio embedding $a_{1:N}$, extracted from the audio $y_{1:N}$ using Hubert (Hsu et al., 2021), implicitly controls the lip motion. Due to the strict alignment with video frames, we apply the audio embedding to its corresponding image. Additionally, the avatar's pose and blink are controlled via explicit signals. The pose is described by a 6D vector Ji et al. (2022). During training, the pose is extracted from video using an open-source tool (Guo et al., 2020). For the blink signal, we adopt the aspect ratio of the left and right eyes, following (Zhang et al., 2023). To account for the arbitrary pose and eye-opening degree of the source image $x_{\text{src}}$, we use the difference between the current frame $x_i$ and the source frame $x_{\text{src}}$: $\Delta\rho_i = \rho_i - \rho_{\text{src}}$, which models the transition of state rather than the state itself. The model is provided with features $z_{\text{src}}$ to supply facial visual details. Each frame's latent code performs cross attention with the audio embedding $a_i$ and pose/blink information $\Delta\rho_i$, respectively. This process injects these conditions into the latent code with different spatial weights, controlling specific regions of the generated content. The image feature $z_{\text{src}}$ is regarded as a global condition and is concatenated directly with the noisy data as the initial input to the 3D U-Net. We also utilize landmarks to create a face region mask for $x_{\text{src}}$, embedded with a lightweight convolutional network, similar to the approach by Tian et al. (2024). This mask adds to the denoising process in the same manner as $z_{\text{src}}$.

**Loss function**. We employed the DM regular denoising loss, $L_{\text{simple}}$, in Equation 2 to train our model. The synchronization of lip motions with audio is crucial for the talking head task, while the lips often constitute only a small portion of the frame. Consequently, during training, we employed landmarks to isolate the lip region by generating a lip mask, $m_{\text{lip}}$. We then applied an additional weight, $w_{\text{lip}}$, to the denoising process of this region, similar to Stypułkowski et al. (2024). Ultimately, our loss function is defined as:

$$L = L_{\text{simple}} + w_{\text{lip}} \cdot (L_{\text{simple}} \otimes m_{\text{lip}}) \tag{7}$$

where the symbol $\otimes$ denotes element-wise product.

### 3.4 Audio-Driven Pose and Blink Generation

To prevent the generated results from exhibiting overly monotonous and minimal movements, we design a separate module, namely the Pose and Blink generation Network (PBNet). As shown in

Figure 1 (b), PBNet learns the mapping between audio and pose/blink movements. To maintain the non-autoregressive (NAR) generation capabilities of A2V-FDM, PBNet employs a transformer-based Variational Autoencoder (VAE) (Petrovich et al., 2021) to generate variable-length pose and blink sequences. The inputs to PBNet include the initial pose/blink state $\boldsymbol{\rho}_{\mathrm{src}}$, the residual pose/blink $\Delta\boldsymbol{\rho}_{1:N}$, and the audio embedding $\boldsymbol{a}_{1:N}$. The transformer encoder $\mathcal{E}_{\mathrm{t}}$ embeds these inputs into a Gaussian-distributed and obtain latent code $\boldsymbol{h} \in \mathbb{R}^{N \times C_h}$ through resampling :

$$\boldsymbol{\mu}, \log\boldsymbol{\sigma} = \mathcal{E}_{\mathrm{t}}(\boldsymbol{\rho}_{\mathrm{src}}, \Delta\boldsymbol{\rho}_{1:N}, \boldsymbol{a}_{1:N}), \ s.t. \ \boldsymbol{h} \sim \mathcal{N}(\boldsymbol{\mu}, \boldsymbol{\sigma}) \tag{8}$$

We design $\boldsymbol{h}$ to have the same length as $\Delta\boldsymbol{\rho}$ to ensure it can carry sufficient information for variable-length inputs. The transformer decoder $\mathcal{D}_{\mathrm{t}}$ generates the final pose/blink sequence $\Delta\hat{\boldsymbol{\rho}}_{1:N}$, conditioned on $\boldsymbol{a}_{1:N}$ and $\boldsymbol{\rho}_{\mathrm{src}}$ :

$$\Delta\hat{\boldsymbol{\rho}}_{1:N} = \mathcal{D}_{\mathrm{t}}(\boldsymbol{h}, \boldsymbol{a}_{1:N}, \boldsymbol{\rho}_{\mathrm{src}}) \tag{9}$$

To enhance the model's extrapolation capability, we use RoPE as the positional encoding in the decoder, consistent with A2V-FDM. During training, we apply an MSE-based reconstruction loss $L_{\mathrm{rec}}$ and an adversarial loss $L_{\mathrm{GAN}}$ to guide the model in completing basic reconstruction tasks (Isola et al., 2016a; Ginosar et al., 2019). Additionally, we employ a KL divergence loss $L_{\mathrm{KL}}$ to ensure that the latent code $\boldsymbol{h}$ closely approximates a standard Gaussian distribution.

### 3.5 TWO-STAGE CURRICULUM LEARNING FOR TALKING HEAD GENERATION

Empirical evidence indicates that training our A2V-FDM model solely with fixed-length short video clips leads to inaccurate control of poses and blinks, as well as poor generalization to longer videos. We argue that a one-step training approach impedes the model's convergence to an optimal solution and fails to achieve satisfactory results in the complex task of talking head generation. To address these issues, we propose an innovative Two-Stage Curriculum Learning (TCL) strategy inspired by the theory of curriculum learning (Bengio et al., 2009).

Overall, the goal of the A2V-FDM during the training process can be expressed as:

$$\hat{\boldsymbol{x}}_{1:N} = \mathcal{D}(\mathrm{DM}(\mathcal{E}(\boldsymbol{x}_{\mathrm{src}}), \boldsymbol{y}_{1:N}, \boldsymbol{\rho}_{1:N})) \tag{10}$$

In the first stage, we set $\boldsymbol{x}_{\mathrm{src}} = \boldsymbol{x}_1$, and the sequence length $N = K'$ is a fixed, relatively small constant. This stage primarily focuses on enabling the model to generate basic lip motions. However, utilizing $\boldsymbol{x}_1$ as the source image often exhibits limited variations in poses and blinks, and using short clips can result in a scarcity of training samples with significant pose or blink movements. Therefore, in the second stage, we set $\boldsymbol{x}_{\mathrm{src}} \in \mathbb{X}$ randomly, where $\mathbb{X}$ is the set of frames in the entire video, to learn control capabilities of large pose transformation. Additionally, differing from stage one, we randomly set $N \in [K_{\min}, K_{\max}], K_{\min} > K'$. This approach aims to enhance control over poses and blinks while maintaining precise lip motions, as longer clips contain more diverse pose and blink movements. Training with random-length sequences also helps the model avoid a bias towards fixed-length sequences, further enhancing the model's extrapolation.

### 3.6 INFERENCE

Our inference process has four steps: 1) Extract the audio embedding $\boldsymbol{a}_{1:N}$. 2) Use the source image $\boldsymbol{x}_{\mathrm{src}}$ to extract the initial pose/blink state $\boldsymbol{\rho}_{\mathrm{src}}$ for PBNet. Along with $\boldsymbol{a}_{1:N}$ and a latent space vector $\boldsymbol{h}_{1:N}$ sampled from a standard Gaussian distribution, PBNet generates the pose/blink sequences $\hat{\boldsymbol{\rho}}_{1:N}$. 3) Input $\boldsymbol{x}_{\mathrm{src}}$, $\boldsymbol{a}_{1:N}$, and $\hat{\boldsymbol{\rho}}_{1:N}$ into the A2V-FDM, which generates the motion representation sequences $\hat{\boldsymbol{z}}_{1:N}^m$. 4) Finally, decode the video sequence $\hat{\boldsymbol{x}}_{1:N}$ from $\boldsymbol{x}_{\mathrm{src}}$ and $\hat{\boldsymbol{z}}_{1:N}^m$. Both PBNet and A2V-FDM generate sequences of dynamic lengths in a single pass, depending on input audio length.

Our method leverages non-autoregressive (NAR) generation during the inference process. To enhance extrapolation during inference, we utilize local attention (Luong, 2015) in the temporal attention module for both the PBNet decoder and the 3D U-Net in A2V-FDM, which restricts the attention scores to a local region. This approach effectively models local dependencies in talking head videos. To accommodate the different temporal dependencies of lip motions and pose/blink movements, we use a larger window size in the local attention mechanism of PBNet compared to A2V-FDM.

## 4 EXPERIMENT

### 4.1 SETUP

**Dataset.** Our method is evaluated on two datasets: CREMA (Cao et al., 2014) and HDTF (Zhang et al., 2021). The CREMA dataset was collected in a controlled laboratory environment and contains 7,442 videos from 91 identities, with durations ranging from 1 to 5 seconds. The HDTF dataset consists of 410 videos, with an average duration exceeding 100 seconds. These videos are gathered from wild scenarios and feature over 10,000 unique sentences for speech content, along with diverse head pose movements. We partitioned the CREMA dataset into training and testing sets following Stypułkowski et al. (2024). As for the HDTF dataset, we conducted a random split with a 9:1 ratio between the training and testing sets. We resized the videos at a resolution of $128 \times 128$ and a frame rate of 25 frames per second (fps) without any additional preprocessing.

**Implementation details.** The architecture of the encoder and decoder in our model aligns with the design proposed by Johnson et al. (2016), while the flow predictor is implemented based on MRAA (Siarohin et al., 2021). The PBNet model is trained using pose and blink movement sequences of 200 frames. During the inference phase of the PBNet model, a local attention mechanism with a window size of 400 is employed. For the inference phase of the A2V-FDM model, local attention with a window size of 80 is applied. In our evaluation, the length of one-time inference for CREMA is dynamic and depends on the ground truth, while for HDTF, it is fixed at 200 frames for better comparison.

**Evaluation metrics.** We evaluate the performance of our method using various metrics. Specifically, we employ the Fréchet Inception Distance (FID) (Heusel et al., 2017) to assess the image quality. We utilize the $FVD_{16}$ and $FVD_{32}$ scores, which calculate the Fréchet Video Distance (FVD) based on window sizes of 16 and 32 frames, respectively, to evaluate video quality across different temporal scales. Furthermore, we assess the perception loss of lip shape using the confidence score ($LSE_C$) and distance score ($LSE_D$) (Chung & Zisserman, 2017). To evaluate the preservation of speaker identity, we use ArcFace (Deng et al., 2019) to extract features from both the ground truth image and the generated image, and use the cosine similarity (CSIM) between the two as the evaluation metric. Moreover, we employ the Beat Align Score (BAS) (Siyao et al., 2022) to evaluate the synchronization of head motion and audio, and calculate the number of blinks per second (blink/s) to assess the liveliness of the eyes. To better illustrate the error accumulation from a quantitative perspective, we design a metric to quantify the severity of error accumulation in the image space inspired by Bian et al. (2022) in the sequential generation task: Degradation Rate (DR), defined as $DR = \frac{FID_{ed}}{FID_{st}} - 1$. DR is related to the ratio between the FID of the last $n$ frames $FID_{ed}$ and the first $n$ frames $FID_{st}$. The motivation for proposing this metric is that when error accumulation occurs, the quality of the generated data at the end of the sequence significantly deteriorates compared to the beginning. A larger DR indicates more severe error accumulation. In our experiments, we set $n$ to 25 frames (1s) and 50 frames (2s), denoted as $DR_{25}$ and $DR_{50}$, respectively.

### 4.2 OVERALL COMPARISON

We compared our method with several state-of-the-art methods: Wav2Lip (Prajwal et al., 2020), Audio2Head (Wang et al., 2021), SadTalker (Zhang et al., 2023), Diffused Heads (Stypułkowski et al., 2024), DreamTalk (Ma et al., 2023), Hallo (Xu et al., 2024b), and EchoMimic (Chen et al., 2024). The quantitative experimental results are shown in Table 1. Our method achieves promising performance in FID, $FVD_{16}$, $FVD_{32}$, CSIM, BAS, and Blink/s metrics for the CREMA and HDTF datasets, as shown in Table 1. It also attains nearly best scores for $LSE_C$ and $LSE_D$. It is important to note that EchoMimic and DreamTalk are applicable only to face-aligned scenarios, whereas other methods do not require pre-cropping. Additionally, Hallo and EchoMimic are built upon pre-trained Stable Diffusion models (Rombach et al., 2022), inheriting substantial visual generation capabilities. Despite this, our method still achieves comparable or even better performance on the HDTF dataset. Furthermore, our method also outperforms Hallo and EchoMimic in terms of generation speed. In Appendix A.4.3, results indicate that our method achieves the fastest or near-fastest generation speed and requires significantly less time than Diffused Heads, Hallo, and EchoMimic.

For the qualitative experiment, as shown in Figure 2, we visualize the generation results for each baseline on different datasets. Our method evidently achieves the visual quality most similar to

Table 1: Quantitative comparison with several state-of-the-art methods methods on HDTF (Zhang et al., 2021) and CREMA (Cao et al., 2014) datasets. We use **bold** to indicate the best score and underline to represent the second-best score. * Wav2Lip generated videos that only contain lip motions, while the rest remain still images. For the sake of rigor, consider it a reference for the quality of lip motion and we will not include Wav2Lip in the ranking. "↑" indicates better performance with higher values, while "↓" indicates better performance with lower values. For both BAS and Blink/s, we consider performance to be better when they are closer to the ground truth.

| | Method | FID↓ | FVD$_{16}$↓ | FVD$_{32}$↓ | LSE$_C$↑ | LSE$_D$↓ | CSIM↑ | BAS | Blink/s |
|---|---|---|---|---|---|---|---|---|---|
| CREMA | GT | - | - | - | 5.88 | 7.87 | 1 | 0.192 | 0.24 |
| | Audio2Head | 29.58 | 188.54 | 208.44 | 5.13 | 7.92 | 0.660 | 0.274 | 0.01 |
| | SadTalker | 16.05 | 101.43 | 158.85 | 5.57 | **7.36** | 0.808 | 0.244 | 0.33 |
| | Diffused Heads | 13.01 | 64.27 | 116.18 | 4.56 | 9.26 | 0.673 | **0.185** | **0.26** |
| | Wav2Lip* | 10.23 | 130.23 | 242.19 | 6.08 | 7.74 | 0.801 | - | - |
| | DAWN (ours) | **5.77** | **56.33** | **75.82** | **5.77** | 8.14 | **0.845** | 0.231 | 0.29 |
| HDTF | GT | - | - | - | 7.95 | 7.33 | 1 | 0.267 | 0.75 |
| | Audio2Head | 30.10 | 122.26 | 205.42 | 6.88 | **7.58** | 0.705 | 0.290 | 0.09 |
| | SadTalker | 26.11 | 97.43 | 187.43 | 6.27 | 8.03 | 0.767 | 0.297 | 0.47 |
| | Wav2Lip* | 23.85 | 166.15 | 281.73 | 7.42 | 7.44 | 0.701 | - | - |
| | DreamTalk | 58.8 | 406.58 | 516.21 | 6.48 | 8.43 | 0.641 | 0.311 | 0.032 |
| | EchoMimic | 32.8 | 139.00 | 178.16 | 6.69 | 8.27 | 0.731 | 0.318 | 0.121 |
| | Hallo | 14.2 | **57.47** | 100.99 | **7.16** | 8.01 | 0.709 | 0.301 | 0.254 |
| | DAWN (ours) | **9.60** | 60.34 | **95.64** | 6.71 | 7.94 | **0.790** | **0.281** | **0.86** |

Table 2: Comparison with other generation strategies. The semi-autoregressive (SAR) generation strategy is similar to He et al. (2023). The two temporal resolution (TTR) generation method is mentioned in Harvey et al. (2022). For the DR metric, we consider performance to be better as it approaches zero.

| Method | Time(s)↓ | FID↓ | FVD$_{16}$↓ | FVD$_{32}$↓ | LSE$_C$↑ | LSE$_D$↓ | DR$_{25}$ | DR$_{50}$ |
|---|---|---|---|---|---|---|---|---|
| SAR | 11.42 | 13.00 | 120.33 | 210.52 | 4.34 | 8.29 | 0.307 | 0.021 |
| TTR | 19.25 | 9.77 | 95.42 | 137.14 | 4.87 | 8.68 | **-0.028** | **-0.005** |
| Ours | **7.32** | **5.77** | **56.33** | **75.82** | **5.77** | **8.14** | 0.044 | 0.031 |

the ground truth, showcasing the most realistic and vivid visual effects. Compared to our method, SadTalker relies on the 3DMM prior, which limits its animation capability to the facial region only, resulting in significant artifacts when merging the face with the static torso below the neck. Additionally, SadTalker exhibits unnatural head pose movements and gaze direction, partially due to limited temporal modeling ability. Audio2Head fails to preserve the speaker's identity during generation. For the HDTF dataset, the Diffused Heads method collapsed due to the error accumulation. DreamTalk is only applicable to face-aligned scenarios, thus requiring the source image to be cropped. Hallo leads to serious error accumulation in the CREMA dataset, resulting in abnormal color patches in the background, indicating a robustness defect.

### 4.3 COMPARISON WITH OTHER GENERATION STRATEGIES

We compared our non-autoregressive generation strategy with two regular video generation strategies: 1) semi-autoregressive (SAR) generation similar to He et al. (2023), and 2) two-temporal resolution (TTR), which trains two models with different temporal resolutions (Harvey et al., 2022). The time cost represents the time required to generate an 8-second talking head video. The models were evaluated on the CREMA dataset, and the results are shown in Table 2. According to the results, our non-autoregressive method produces videos with the highest overall quality and fastest speed. In addition, the SAR method results in a very high DR, which is at least one order of magnitude higher

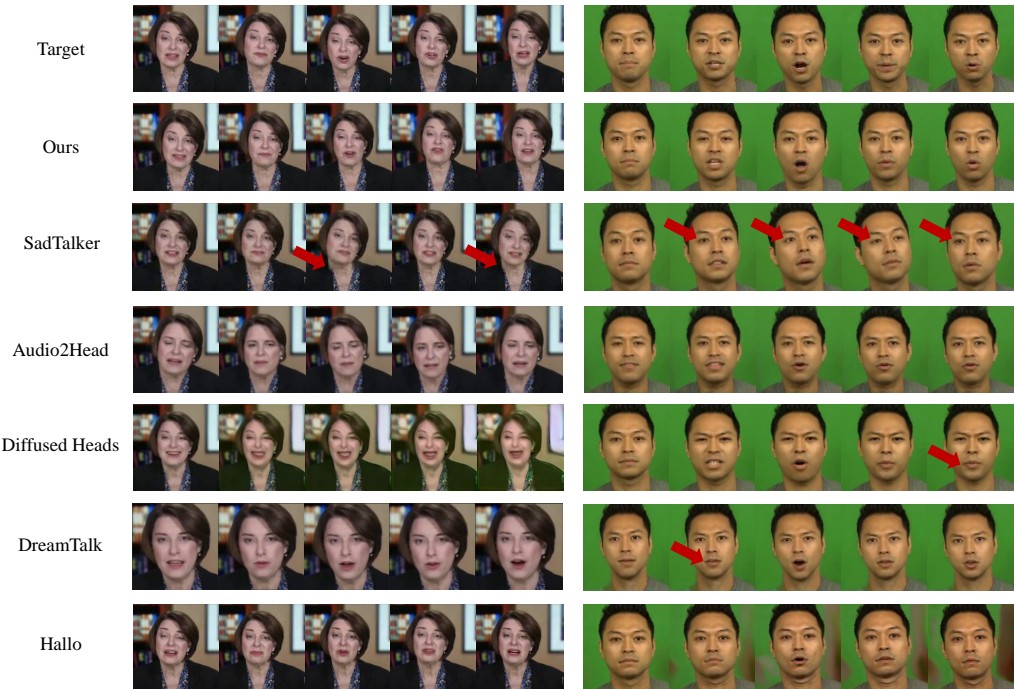

Figure 2: Qualitative comparison with several state-of-the-art methods methods on HDTF (Zhang et al., 2021) and CREMA (Cao et al., 2014) datasets. Our method produces higher-quality results in video quality, lip-sync consistency, identity preservation, and head motions.

Table 3: The experiment of extrapolation evaluation. "Inference length" refers to the number of frames generated in a single inference process.

| Inference length | FID↓ | FVD$_{16}$↓ | FVD$_{32}$↓ | LSE$_C$↑ | LSE$_D$↓ |
|---|---|---|---|---|---|
| 40 | 9.35 | 59.58 | 94.09 | 5.76 | 7.89 |
| 100 | 9.83 | 61.72 | 98.80 | 6.41 | 7.96 |
| 200 | 9.60 | 60.34 | 95.64 | 6.71 | 7.94 |
| 400 | 10.36 | 61.57 | 97.84 | 6.63 | 8.12 |
| 600 | 10.30 | 60.44 | 96.62 | 6.76 | 8.02 |

than TTR and NAR. This is because autoregressive methods suffer from degradation issues during iterative generation. Although TTR also successfully alleviates the degradation issue, it compromises generation speed and overall quality. In summary, our non-autoregressive method addresses the degradation problem while preserving fast generation speed and high overall quality.

## 4.4 EXTRAPOLATION VALIDATION

To evaluate the extrapolation ability of our method, we used the HDTF dataset to assess the impact of inference length on model performance, ranging from 40 to 600 frames. FID and FVD metrics remain stable with inference length, while longer audio improves lip movement precision. This suggests that inference with sufficient length helps the model produce more precise lip movement. Furthermore, to better compare the extrapolation of other AR-based DM methods, Hallo was selected for comparison. As shown in Table 5 from the Appendix A.3, our method's DR metric stabilizes within a certain range, in contrast to Hallo's error accumulation at increased inference lengths.

Table 4: Ablation study on TCL and PBNet. The "GT PB" refers to whether to use ground truth pose/blink signal.

| Method | GT PB | FID↓ | FVD$_{16}$↓ | FVD$_{32}$↓ | LSE$_C$↑ | LSE$_D$↓ |
|---|---|---|---|---|---|---|
| only stage 1 | ✓ | **7.95** | 81.84 | 126.52 | 4.38 | 10.04 |
| only stage 2 | ✓ | 13.71 | 125.75 | 166.83 | 6.14 | 8.43 |
| DAWN | ✓ | 9.68 | **52.05** | **87.11** | **6.71** | **7.99** |
| w/o PBNet | ✗ | 15.20 | 100.94 | 162.35 | 5.79 | 8.36 |
| DAWN | ✗ | **9.60** | **60.34** | **95.64** | **6.71** | **7.94** |

## 4.5 ABLATION STUDY

**Ablation study on TCL strategy.** Since TCL strategy is used only on A2V-FDM, PBNet is excluded by using ground truth pose/blink for evaluation. In this section, we extend the number of training epochs based on data throughput when using single-stage training. A2V-FDM trained with either stage 1 or 2 separately shows that stage 1 decreases overall performance except for FID, due to shorter clip training causing minor warping and lower FID scores but poorer FVD. Stage 2 improves LSE$_C$ and LSE$_D$ but results in worse FID and FVD as the model struggles with simultaneous pose/blink control and lip movement. Both stages alone underperform compared to the proposed TCL strategy, underscoring its effectiveness.

**Ablation study on PBNet.** We evaluate the effectiveness of the PBNet in Table 4. The term "w/o PBNet" indicates that the PBNet module was removed from the architecture, requiring the A2V-FDM to simultaneously generate pose, blink, and lip motions from the audio by itself. The results suggest an overall enhancement of all evaluation metrics with the inclusion of PBNet. This is because modeling the long-term dependency of pose and blink movements through PBNet simplifies training for the A2V-FDM. We also visualized the effectiveness of PBNet in Appendix A.4.5.

## 5 CONCLUSION

We introduce DAWN, an innovative architecture that non-autoregressively generates dynamic frames of talking head videos from given portraits and audio. We utilized the LFG to extract motion representations from speech videos. To produce vivid talking head videos, we propose PBNet and A2V-LDM. The PBNet generates natural pose/blink movements from speech, while A2V-LDM produces motion representations conditioned on audio and pose/blink movements. Finally, these generated motion representations are decoded into videos using LFG. We demonstrate on two datasets that our model can generate extended talking head videos with high-quality dynamic frames in a single pass, achieving realistic visual effects, accurate lip synchronization, and strong extrapolation capabilities.

## 6 LIMITATION AND FUTURE WORKS

Our work still has certain limitations. For instance, the model cannot fully comprehend physical common sense during generation, particularly when individuals in portraits are wearing items such as hats, helmets, or headpieces. Sometimes, these items do not move with the head, causing artifacts in the results. We aim to inject these physical dependencies into the model without sacrificing vividness in future work. Additionally, our architecture currently requires each sub-module to be trained separately. In future work, we plan to enable joint training to reduce error propagation across modules.

ACKNOWLEDGMENTS

This work was supported by the National Natural Science Foundation of China under Grant 62171427.

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

# A APPENDIX

## A.1 SOCIAL IMPACT CONSIDERATION

DAWN focuses on creating realistic talking head videos with the aim of generating positive social impact. We firmly oppose the malicious misuse of our method, including fraud, creating fake news, and violating portrait rights. Thus, we assert that our open-source code and model should be used exclusively for research purposes. We hope that our technique can provide social benefits in the future, such as promoting education, enabling face-to-face communication for separated people, and treating certain psychological disorders.

## A.2 TRAINING DETAILS

Since the HDTF dataset contains fewer videos of longer length compared to the CREMA dataset, each video in the HDTF dataset is sampled 25 times per epoch to achieve better I/O performance. The LFG is fine-tuned independently on the training datasets of CREMA and HDTF, using a checkpoint pre-trained on the VoxCeleb (Nagrani et al., 2017) dataset. We fine-tune the LFG with 400 epochs. During the training of A2V-FDM, we trained with 500 epochs for each stage. In both stages, we select the $w_{\text{lip}} = 1$ for the loss function in Equation 7. We freeze the parameters of LFG when training the A2V-FDM. Although end-to-end training can prevent errors in the LFG from affecting the A2V-FDM, the main reason for not adopting end-to-end training is the difficulty in achieving stable convergence. In an end-to-end training setup, the A2V-FDM is likely to receive incorrect supervision signals, which can hinder the training process of the model and make it highly unstable. For PBNet, the MSE-based reconstruction loss can be expressed as:

$$\mathcal{L}_{\text{MSE}} = \frac{1}{T} \sum_{n=1}^{N} (\Delta\hat{\boldsymbol{\rho}}_n - \Delta\boldsymbol{\rho}_n)^2 \tag{11}$$

where $\Delta\hat{\boldsymbol{\rho}}$ is the predicted pose/blink sequences, and $\Delta\boldsymbol{\rho}$ is the ground truth sequences. We also implement a KL divergence loss to constrain the latent code $\boldsymbol{h}$ closely approximates a standard Gaussian distribution. The adversarial loss $L_{\text{GAN}}$ is defined as:

$$L_{GAN} = \arg\min_{G} \max_{D} (G, D) \tag{12}$$

where the $G$ is our proposed PBNet and the $D$ is a discriminator implemented based on PatchGAN (Isola et al., 2016b). We use a 1D convolution based network to output the probability of each patch in the pose/blink sequence originating from real actions. The discriminator is guided by the BCE loss. In the training, the loss function of PBNet is set as:

$$L = \lambda_{\text{rec}} L_{\text{rec}} + \lambda_{\text{KL}} L_{\text{KL}} + \lambda_{\text{GAN}} L_{\text{GAN}} \tag{13}$$

where we set $\lambda_{\text{rec}} = 1$, $\lambda_{\text{GAN}} = 0.6$. During the training process, we observed that $L_{\text{KL}}$ rapidly converges to approximately zero at the beginning. This convergence impairs the latent state's ability to retain effective information, causing the model to depend almost entirely on the decoder's predictive capability for completing the fixed fitting task from audio to pose/blink. To ensure diversity in generation, we implemented a method of progressively increasing the $\lambda_{\text{KL}}$ in training PBNet. The PBNet was trained over 1600 epochs. In the initial 400 epochs, we did not apply the $L_{\text{KL}}$ constraint to the latent code., which helped the model develop basic sequence reconstruction capabilities in the early training phase. From that point until the end of training, $\lambda_{\text{KL}}$ was gradually increased to 0.01.

## A.3 ADDITIONAL EXPERIMENTS

### A.3.1 EXTRAPOLATION COMPARISON

To highlight the advantages of NAR generation, we selected an image-based diffusion method, Hallo, for comparison, due to its representative structure similar to most of the latest AR or SAR methods. We evaluated the degradation rate for both our model and Hallo across inference lengths of 200, 400, and 600, as detailed in Table 5. The results indicate that, in our method, both the $\text{DR}_{25}$ and $\text{DR}_{50}$ metrics remain stable as inference length increases. In contrast, Hallo's $\text{DR}_{25}$ and $\text{DR}_{50}$ metrics significantly increase with longer inference lengths, suggesting notable error accumulation in the

| Source
Image | Generated
Frames | Source
Image | Generated
Frames |

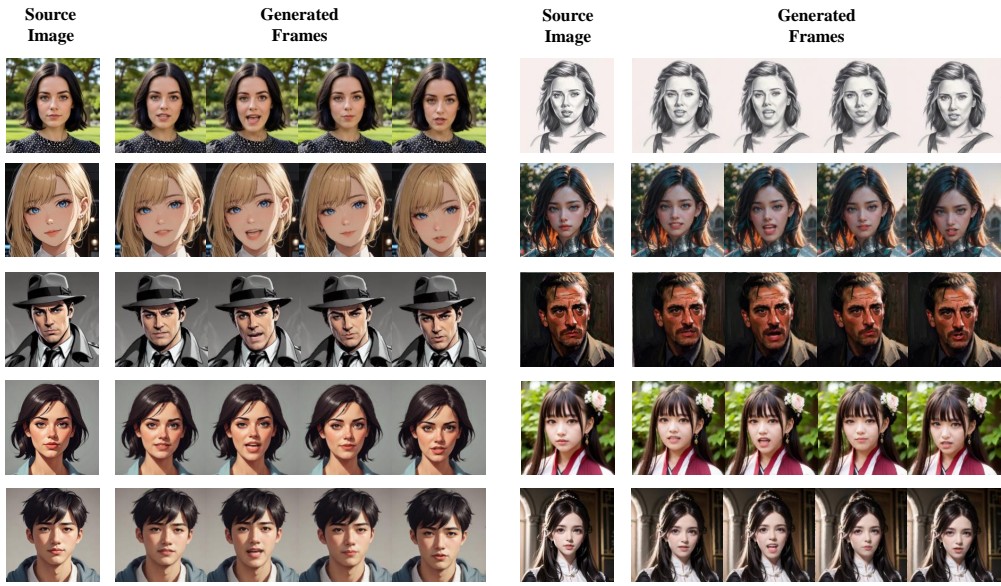

Figure 3: The qualitative study on higher resolution ($256 \times 256$) and different portrait styles.

Table 5: The comparison experiment of error accumulation with Hallo. "Inference length" refers to the number of frames generated in a single inference process.

| Inference length | DAWN | | Hallo | |
|---|---|---|---|---|
| | $DR_{25}$ | $DR_{50}$ | $DR_{25}$ | $DR_{50}$ |
| 200 | 0.253 | 0.043 | 0.214 | 0.094 |
| 400 | 0.208 | 0.164 | 0.279 | 0.161 |
| 600 | 0.152 | 0.152 | 0.422 | 0.332 |

image space. This demonstrates that image-based AR methods, such as Hallo, struggle with error accumulation in longer tasks, while our model exhibits superior extrapolation capabilities and more consistent performance with extended video sequences. These findings underscore the superiority of NAR methods compared to AR and SAR alternatives.

## A.4 USER STUDY

The user study evaluates generated videos across four dimensions: 1) L-Sync: Lip-audio synchronization; 2) O-Nat: The overall naturalness of the generated talking head ; 3) M-Viv: The vividness of the head movements; 4) V-Qual: The overall video quality (e.g., presence of artifacts or abnormal color blocks). We generated 10 test videos per method, with 23 participants scoring each on a 1-5 scale. Users disregarded resolution and cropping when rating. As shown in Table 6, our method surpasses existing approaches in lip synchronization, naturalness, and head movement vividness, and is comparable to EchoMimic in video quality.

### A.4.1 POSE/BLINK CONTROLLABLE GENERATION

In addition to generating lifelike avatars, our method also enables the controllable generation of pose and blink actions. Users can either use pose and blink information generated by our PBNet or provide these sequences directly, such as by extracting them from a given video. The results, as shown in Figure 4, demonstrate that our method not only provides high-precision control over the pose/blink

Table 6: User study.

| Method | L-Sync | O-Nat | M-Viv | V-Qual |
|---|---|---|---|---|
| Audio2Head | 3.87 | 3.67 | 3.28 | 3.66 |
| Sadtalker | 4.23 | 3.13 | 2.81 | 4.14 |
| DreamTalk | 4.38 | 3.89 | 3.41 | 4.42 |
| Hallo | 4.40 | 3.76 | 3.89 | 4.03 |
| EchoMimic | 4.45 | 4.30 | 4.06 | **4.53** |
| DAWN(ours) | **4.57** | **4.41** | **4.43** | 4.51 |

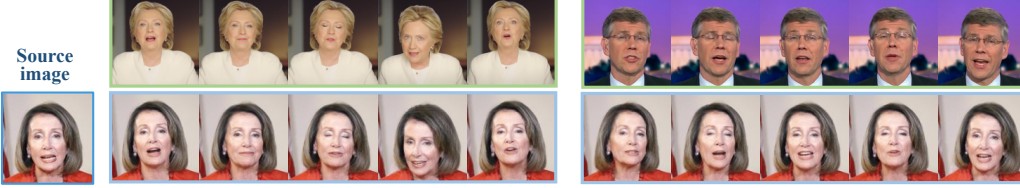

Figure 4: Visualization of cross-identity reenactment. We extract the audio, head pose, and blink signals from the video in the first row, and use them to drive the source image, generating the talking head video in the second row.

movements of the generated avatars, but also effectively transfers rich facial dynamics, including expressions, blinks, and lip motions.

### A.4.2 EXPERIMENT ON HIGHER RESOLUTION AND DIFFERENT PORTRAIT STYLES

We further investigate the generalization ability of our method on higher-resolution images and different portrait styles. We trained DAWN on the HDTF dataset with a resolution of $256 \times 256$. Then we evaluated our higher-resolution model on the HDTF test set. The quantitative results are illustrated in Table 7. The results show that our method still maintains strong competitiveness compared to the latest talking head models. We also test our method on multiple out-of-dataset source images featuring diverse styles, as showcased in Figure 3. The results indicate that our method yields promising outcomes in high-resolution generation and demonstrates considerable generalization ability across various image styles, including photos, paintings, anime, and sketches.

### A.4.3 COMPARISON EXPERIMENT ON GENERATION TIME COST

We experimented to test the time cost of video generation. We generated 8-second talking head videos with the same source image and audio, then recorded the time consumption for each method. To ensure a fair comparison, we excluded the audio encoding step for all methods. The testing was performed on a single V100 16G GPU. As shown in Figure 5, our method achieves the fastest or near-fastest generation speed and requires significantly less time compared to the previous diffusion-based methods, Diffused Heads, Hallo, and EchoMimic.

### A.4.4 ABLATION STUDY ON THE LOCAL ATTENTION MECHANISM

In our work, we utilized a local attention mechanism to enhance the extrapolation capability of our model. We conducted experiments to evaluate the effect of varying the window size of the local attention mechanism in A2V-FDM, ranging from 20 to 200, and also assessed the model's performance without the local attention mechanism. To eliminate the influence of the generated pose/blink, we used the ground-truth pose/blink signals to drive the model. The results, presented in Table 8, indicate that the model's performance peaks around a window size of 80. This is attributed

Table 7: Quantitative study on different resolutions. The "128" indicates our method is trained on a $128 \times 128$ resolution, and the "256" indicates training on a $256 \times 256$ resolution.

| Method | FID↓ | FVD$_{16}$↓ | FVD$_{32}$↓ | LSE$_C$↑ | LSE$_D$↓ | CSIM↑ | BAS | Blink/s |
|--------|------|-------------|-------------|----------|----------|-------|-----|---------|
| GT | - | - | - | 7.95 | 7.33 | 1 | 0.267 | 0.75 |
| DAWN (128) | 9.60 | 60.34 | 95.64 | 6.71 | 7.94 | 0.790 | 0.281 | 0.86 |
| DAWN (256) | 11.80 | 68.07 | 105.20 | 7.20 | 7.80 | 0.791 | 0.278 | 0.73 |

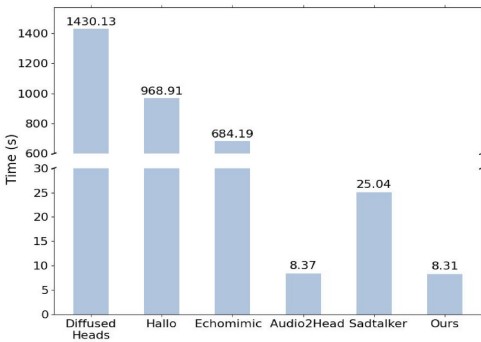

Figure 5: The comparison experiment on generation time cost. The Diffused Heads, Hallo, EchoMimic are existing diffusion-based methods.

to the maximum length of the video clips during training, which is 40 frames. Consequently, the model learns to process temporal dependencies within a 40-frame distance. Thus, using the window size of twice the max training clip length takes full advantage of the model's capability. Reducing or increasing the window size degrades performance; a smaller window size leads to a loss of contextual information, significantly impairing the model's performance, while a larger window size or the absence of the local attention mechanism reduces extrapolation ability, also resulting in lower performance. Since extrapolation ability is also supported by our TCL strategy and the intrinsic structure of A2V-FDM, removing the local attention mechanism causes relatively minor damage compared to the loss of contextual information.

### A.4.5 ABLATION STUDY ON THE PBNET

We further provide the visualization of the ablation study on PBNet in Figure 6, while the quantitative results are illustrated in Table 4. It suggests that using the PBNet to generate the pose exclusively will provide the pose with more vividness and diversity. However, generating lip, head pose, and blink movement from audio simultaneously will cause a relatively static head pose, which severely impacts the vividness and naturalness.

Table 8: Ablation study on the local attention mechanism. The "window" means the window size in the local attention operation. The "None" means we use the original attention mechanism instead.

| Window | FID$\downarrow$ | FVD$_{16}\downarrow$ | FVD$_{32}\downarrow$ | LSE$_C\uparrow$ | LSE$_D\downarrow$ |
|--------|------|---------|---------|--------|--------|
| 20 | 14.47 | 159.19 | 217.54 | 5.69 | 8.97 |
| 40 | 10.93 | 72.93 | 114.52 | 6.35 | 8.33 |
| 80 | 9.68 | **52.05** | **87.11** | **6.71** | 7.99 |
| 200 | **9.44** | 53.48 | 88.84 | 6.60 | **7.94** |
| None | 9.70 | 63.95 | 103.83 | 6.37 | 8.15 |

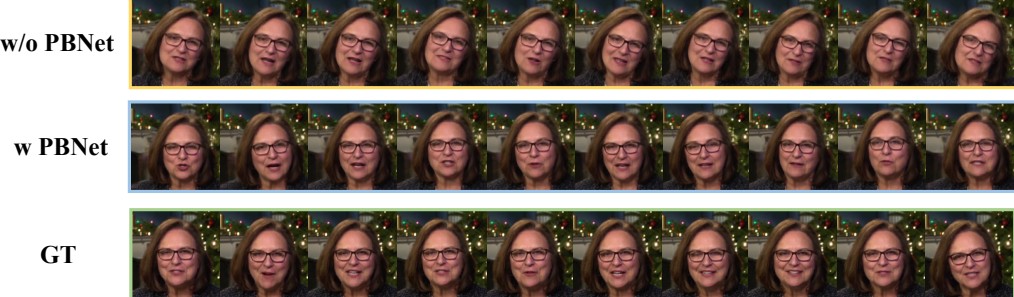

Figure 6: The visualization of the ablation study on PBNet demonstrates different methodologies. The term "w/o PBNet" indicates "without PBNet", whereby the A2V-FDM is utilized to infer pose and blink movements. Conversely, "w PBNet" signifies the "with PBNet", which directly generates explicit pose and blink signals to control the generation of A2V-FDM.

