# OpenReview forum: "DAWN: Dynamic Frame Avatar with Non-autoregressive Diffusion Framework for Talking head Video Generation"
_ICLR.cc/2025/Conference — ICLR 2025 Poster_

### Official Review · Reviewer_5zi1 · 2024-10-18

**Soundness:** 3
**Presentation:** 3
**Contribution:** 2
**Rating:** 5
**Confidence:** 5

**Summary:**

The paper introduces DAWN, a system that uses non-autoregressive (NAR) diffusion models to generate talking head videos of varying lengths from portrait images and audio. This method achieves faster processing speeds and high-quality outputs. To overcome limitations in NAR approaches and improve the modeling of longer videos, the system separately controls the movements of the lips, head, and blinking. This provides more precise control over these individual movements. The paper proposes PBNet, a network that generates realistic head poses and blinking sequences directly from audio clips using an NAR approach. The Two-stage Curriculum Learning (TCL) strategy is introduced to train the model effectively in generating lip movements and controlling head poses and blinks accurately, which helps in achieving robust convergence and better extrapolation capabilities.

**Strengths:**

1. The performance is decent
2. Talking head generation is a research field with practical value.

**Weaknesses:**

1. The motivation for introducing NAR is relatively weak. VASA[1], which uses SAR, already achieves real-time performance with good results. The motivation to use NAR to improve inference speed is weak, as the paper is only faster than diffused head, but the slower speed of diffused head is due to it generating images directly rather than intermediate representations.
2. The novelty of this paper is limited, as many modules are similar to previous methods. For example, the LFG is similar to the FOMM[2] series of work, and pose prediction is similar to SadTalker[3]. Although the paper introduces some novel training techniques like TCL, previous methods have already achieved good results without relying on such fancy techniques. Therefore, the significance of these fancy techniques is questionable.

[1] Xu, Sicheng, et al. "Vasa-1: Lifelike audio-driven talking faces generated in real time." arXiv preprint arXiv:2404.10667 (2024).

[2] Siarohin, Aliaksandr, et al. "First order motion model for image animation." Advances in neural information processing systems 32 (2019).

[3] Zhang, Wenxuan, et al. "Sadtalker: Learning realistic 3d motion coefficients for stylized audio-driven single image talking face animation." Proceedings of the IEEE/CVF Conference on Computer Vision and Pattern Recognition. 2023.

**Questions:**

Did the paper conduct a user study? I don’t seem to have seen one. User studies are very important in this field.

---

> ### Author Response · Authors · 2024-11-17
> **Author Response to Official Review by Reviewer 5zi1 (1/4)**
>
> Dear reviewer 5zi1, thank you for taking the time to review our paper and provide valuable suggestions. We hope our following comments address your concerns. Additionally, we have uploaded new supplementary videos to the [anonymous repository](https://anonymous.4open.science/r/DAWN-BC9F/README.md). Since the files are large, for a better viewing experience, we recommend downloading the supplementary materials first.
>
>
> > Q1：The motivation for introducing NAR is relatively weak. VASA, which uses SAR, already achieves real-time performance with good results. The motivation to use NAR to improve inference speed is weak, as the paper is only faster than diffused head, but the slower speed of diffused head is due to it generating images directly rather than intermediate representations.
>
>
> Although VASA has achieved promising results, it heavily relies on 3D representation [1]. This reliance may result in artifacts when VASA processes characters where standard human 3D representation is challenging, such as in cartoons or anime [1]. In contrast, our approach relies much less on 3D representation, thereby enhancing its generalization ability. This attribute also has received positive feedback from reviewer Q1nh. Additionally, we have provided new supplementary materials covering out-of-dataset test samples across various character types and multiple languages, further demonstrating our approach's versatility (filename: [Chinese.mp4](https://anonymous.4open.science/r/DAWN-BC9F/Chinese.mp4), [French.mp4](https://anonymous.4open.science/r/DAWN-BC9F/French.mp4), [German.mp4](https://anonymous.4open.science/r/DAWN-BC9F/German.mp4), [Japanese.mp4](https://anonymous.4open.science/r/DAWN-BC9F/Japanese.mp4), [Korean.mp4](https://anonymous.4open.science/r/DAWN-BC9F/Korean.mp4) ).
>
> Thank you for your suggestions. At present, AR or SAR methods have indeed achieved good results. They have somewhat alleviated difficulties such as slow inference speed, error accumulation through certain techniques. However, we hope to explore the potential of NAR methods to avoid these risks more effectively. Since VASA is not open source, for the sake of rigor, we have added an **overall generation speed comparison** of Hallo[3]  and Echomimic[4], since both of them use diffusion models to generate intermediate representations of the image. We have additionally included **quantitative comparison** experiments with DreamTalk [2], Hallo, and EchoMimic. The results are shown below. It is important to note that EchoMimic and DreamTalk are only applicable to face-aligned scenarios, whereas other methods do not require pre-cropping. Additionally, Hallo and EchoMimic are built upon pre-trained Stable Diffusion models, inheriting substantial visual generation capabilities. Despite this, our method still achieves comparable or even better performance on the HDTF dataset. Furthermore, our method also outperforms Hallo and EchoMimic in terms of generation speed.
>
>
> | Method | Inference time (s)  |
> |----------|----------|
> | DiffusedHeads| 1430.13     |
> | Echomimic    | 684.19     |
> | Hallo        | 968.91     |
> | DAWN(ours)   | 8.31     |
>
>
> |          | FID      | FVD16 |FVD32 |LSE-C | LSE-D | CSIM     | BAS      | Blink/s  |
> |----------|----------|----------|----------|----------|----------|----------|----------|----------|
> | GT        | -        | -        | -        | 7.95     | 7.33     | 1       | 0.267     | 0.75     |
> | DreamTalk | 58.8     | 406.58     | 516.21     | 6.48     | 8.43     | 0.641     | 0.3110     | 0.032     |
> | Hallo     | 14.2     | **57.47**     | 100.99     | **7.16**     | 8.01     | 0.7095     | 0.3010     | 0.254     |
> | Echomimic | 32.8     | 139.00     | 178.16     | 6.69     | 8.27     | 0.7314     | 0.3177     | 0.1212     |
> | DAWN(ours)    | **9.60**    | 60.34     | **95.64**    | 6.71     | **7.94**    | **0.790**     | **0.281**     | **0.86**     |
>
> \
> \
> [1] Xu, Sicheng, et al. "Vasa-1: Lifelike audio-driven talking faces generated in real time." arXiv preprint arXiv:2404.10667 (2024).
>
> [2] Ma, Yifeng, et al. "Dreamtalk: When expressive talking head generation meets diffusion probabilistic models." arXiv preprint arXiv:2312.09767 (2023).
>
> [3] Xu, Mingwang, et al. "Hallo: Hierarchical Audio-Driven Visual Synthesis for Portrait Image Animation." arXiv preprint arXiv:2406.08801 (2024).
>
> [4] Chen, Zhiyuan, et al. "Echomimic: Lifelike audio-driven portrait animations through editable landmark conditions." arXiv preprint arXiv:2407.08136 (2024).
> \
> \
> Due to character limitations, please refer to the next comment for other questions.

---

> ### Author Response · Authors · 2024-11-17
> **Author Response to Official Review by Reviewer 5zi1 (2/4)**
>
> In this comment, we continue to answer Q1.
> \
> \
> One of our major motivations for introducing NAR is to effectively address error accumulation, which can be observed at [Compare_with_Hallo.mp4](https://anonymous.4open.science/r/DAWN-BC9F/Compare_with_Hallo.mp4). To better illustrate the error accumulation from a quantitative perspective, we design a metric to quantify the severity of error accumulation inspired by the work[5] in sequential generation: **Degradation Rate (DR)**, defined as: $\text{DR} = \frac{\text{FID}\_{\text{ed}}}{\text{FID}\_{\text{st}}} -1$.
>
> The motivation for proposing this metric is that when error accumulation occurs, the quality of the generated data at the end of the sequence significantly deteriorates compared to the beginning. DR is related to the ratio between the Fréchet Inception Distance (FID) of the last $n$ frames $\text{FID}\_{\text{ed}}$ and the first $ n$ frames ​$\text{FID}\_{\text{st}}$. A larger DR indicates more severe error accumulation. In our experiments, we set $n$ to 25 frames (1s) and 50 frames (2s), denoted as $\text{DR}\_{25}$ and $\text{DR}\_{50}​$, respectively. Additionally, we selected Hallo[3] for comparison, as it represents most AR or SAR diffusion model structures and has significant influence in the community.
>
> **$\text{DR}\_{25}​$:**
>
> | Inference length | DAWN  | Hallo  |
> |----------|----------|----------|
> | 200    | 0.253     | 0.214     |
> | 400    | 0.208     | 0.279     |
> | 600    | 0.152     | 0.422     |
>
> **$\text{DR}\_{50}​$:**
>
> | Inference length | DAWN  | Hallo  |
> |----------|----------|----------|
> | 200    | 0.043     | 0.094     |
> | 400    | 0.164     | 0.161     |
> | 600    | 0.144     | 0.332     |
>
> It can be observed that the DR index increases progressively with the inference length for Hallo. However, the DR metric of our method **stabilizes within a certain range**. This indicates that Hallo experiences significant error accumulation as the inference length increases, whereas our performance remains stable.
>
> Additionally, we have added a comparison of generation strategies in Table 2. It can be observed that the SAR inference strategy shows significant error accumulation, while our method exhibits minimal error accumulation, thanks to the NAR generation. Although TTR can also avoid error accumulation, it does not perform as well as the NAR approach in terms of generation speed and quality. These experiments further reinforce the motivation behind our proposed NAR approach. We will include the supplementary experiment related to DR in Tables 2 and 3 of the paper.
>
> | Method   | $\text{DR}\_{25}​$ | $\text{DR}\_{50}​$ |
> |----------|----------|----------|
> | SAR    | 0.307     | 0.201    |
> | TTR    | -0.028   | -0.005  |
> | Ours   | 0.044     | 0.031    |
>
> Moreover, when the number of frames to be inferred at the end of a video is less than that of a single inference, SAR requires additional boundary handling. We believe that using NAR can directly and effectively avoid these issues without extra processing.
> \
> \
> \
> [5] Bian, Xiaohang, et al. "Handwritten mathematical expression recognition via attention aggregation based bi-directional mutual learning." Proceedings of the AAAI conference on artificial intelligence. Vol. 36. No. 1. 2022.

---

> ### Author Response · Authors · 2024-11-17
> **Author Response to Official Review by Reviewer 5zi1 (3/4)**
>
> > Q2: The novelty of this paper is limited, as many modules are similar to previous methods. For example, the LFG is similar to the FOMM series of work, and pose prediction is similar to SadTalker. Although the paper introduces some novel training techniques like TCL, previous methods have already achieved good results without relying on such fancy techniques. Therefore, the significance of these fancy techniques is questionable.
>
> Our main innovation is the proposal of a fully non-autoregressive generative framework, where both the A2V-FDM and PBNet support the one-time generation of dynamic frames. Although LFG is similar to FOMM, we chose it for motion representation primarily due to its effectiveness. We also chose a similar SOTA video-driven method LIA[6] to compare with our LFG. The "LIA" and "LFG" mean these two methods pretrained on Voxceleb[9] dataset, and "LFG-finetune" means the LFG fine-tuned on HDTF training set. The results demonstrate the effectiveness of LFG.
>
> | Method  |   FID      |FVD16|FVD32 |LSE-C | LSE-D |
> |----------|----------|----------|----------|----------|----------|
> | LIA    |26.64     | 95.75   | 178.62   | 6.69     | 8.12     |
> | LFG    |12.69    |55.24      | 105.82   | 6.72  | 8.32  |
> | LFG-finetune    |7.23    |24.80      | 38.00   | 7.36     | 7.73     |
>
>
> Although our motivation for using VAE in the pose prediction task is somewhat similar to that of SadTalker[7], our approach to modeling pose/blink motion sequences differs significantly. Our goal is to generate a motion sequence with a dynamic number of frames in a single pass, demonstrating strong extrapolation capabilities. In contrast, SadTalker uses the MLP to produce 32-frame action block for single generation, and achieves long-sequence generation by directly stacking them. This leads to poor contextual utilization, such as abrupt gaze directions, as mentioned in the limitation section of [7]. In terms of performance, we achieve more natural and vivid head movements compared to SadTalker. Reviewer Q1nh also acknowledges our impressive head pose movements in ["driven_by_music.mp4"](https://anonymous.4open.science/r/DAWN-BC9F/Driven_by_music.mp4) from our supplement material.
>
> The introduction of TCL aims to enhance the extrapolation performance of NAR. Most AR-based models design various forms of **reference nets** which produce the AR prior from previous frames to ensure the extrapolation ability, as mentioned in [1,3,4,8] (reference net has nearly 1G of parameters, whereas our entire model is less than 300M). However, our NAR model avoids the introduction of reference nets by using the TCL training strategy to achieve longer inference length (600 frames) under short-frame training conditions (< 40 frames). The effectiveness of TCL is also supported by our experiment in Table 4. We attempt to propose TCL as a more effective training strategy to replace the introduction of reference nets in existing work, thereby enhancing the model's extrapolation capabilities. The concept of TCL is simple yet effective, without complicating the network structure. Therefore, we believe the introduction of TCL is meaningful. We hope that the above discussion can better elucidate the novelty of our work and address your concerns.
> \
> \
> \
> [6] Wang, Yaohui et al. “LIA: Latent Image Animator.” IEEE Transactions on Pattern Analysis and Machine Intelligence 46 (2024): 10829-10844.
>
>
> [7] Zhang, Wenxuan, et al. "Sadtalker: Learning realistic 3d motion coefficients for stylized audio-driven single image talking face animation." Proceedings of the IEEE/CVF Conference on Computer Vision and Pattern Recognition. 2023.
>
> [8] Tian, Linrui, et al. "Emo: Emote portrait alive-generating expressive portrait videos with audio2video diffusion model under weak conditions." arXiv preprint arXiv:2402.17485 (2024).
>
> [9] Nagrani, Arsha et al. “VoxCeleb: A Large-Scale Speaker Identification Dataset.” Interspeech (2017).

---

> ### Author Response · Authors · 2024-11-17
> **Author Response to Official Review by Reviewer 5zi1 (4/4)**
>
> > Q3: Did the paper conduct a user study? I don’t seem to have seen one. User studies are very important in this field.
>
> Thank you for your suggestion. We recognize that human evaluation is indeed indispensable, so we conducted the **user study**. The study evaluates the generated videos from four dimensions: 1) **L-Sync**: Lip-audio synchronization; 2) **O-Nat**: The overall naturalness of the generated talking head ; 3) **M-Viv**: The vividness of the head movements; 4) **V-Qual**: The overall video quality (e.g., presence of artifacts or abnormal color blocks). We generated 10 test videos for each method, and a total of 23 participants took part in the user study, scoring each dimension on a scale of 1 to 5. To strengthen the comparison, we added several recent diffusion-based  talking head methods to the original baselines, including **DreamTalk**, **Hallo**, and **EchoMimic**. Since these methods vary in resolution and some employ image cropping to centralize the face [2,4], we instructed users to disregard resolution and cropping during testing when providing their ratings.
>
>
> As shown in the results, our method outperforms existing approaches in lip synchronization, overall naturalness, and head movement vividness. In terms of video quality, our approach is comparable to the concurrent work EchoMimic.
> It is worth noting that although Hallo achieved high FID/FVD scores in the comparative study, it received lower scores in human evaluations. This is because humans are particularly sensitive to quality degradation resulting from error accumulation, which has been visualized in our supplement material ([Compare_with_Hallo.mp4](https://anonymous.4open.science/r/DAWN-BC9F/Compare_with_Hallo.mp4)). We will include the user study as a section in the experiment part of the paper.
>
> |           | L-Sync | O-Nat | M-Viv |V-Qual |
> |----------|----------|----------|----------|----------|
> | Audio2Head| 3.87     | 3.67     |3.28      |3.66      |
> | Sadtalker| 4.23     | 3.13     |2.81      |4.14      |
> | DreamTalk   | 4.38     | 3.89     |3.41      |4.42      |
> | Hallo   | 4.40     | 3.76     |3.89      |4.03      |
> | Echomimic   | 4.45     | 4.30     |4.06      |**4.53**      |
> | DAWN(ours)   | **4.57**     | **4.41**     |**4.43**      |4.51      |

---

> ### Author Response · Authors · 2024-11-22
> **We hope that our response addresses your concern**
>
> Dear Reviewer 5zi1,
>
> We greatly appreciate the time you've invested in reviewing our response. Having submitted our rebuttal, we are eager to know if our response has addressed your concern. As the end of the rebuttal phase is approaching, we look forward to hearing from you for any further clarification that you might require.
>
> Best,
>
> Submission 6731 authors

---

> > ### Comment · Reviewer_5zi1 · 2024-11-23
> >
> > Thanks for the rebuttal. However, my concerns are not fully addressed.
> > I deeply understand that research in the talking head domain entails a substantial workload, and the authors’ efforts are indeed very arduous. Furthermore, the authors have provided very detailed responses to my questions, for which I am sincerely grateful. However, if I may speak frankly, the responses further indicate that the contributions of this paper are incremental.
> > VASA has already achieved real-time performance with satisfactory results, yet the authors insist on using NAR to address an issue that has already been resolved. The authors insist on highlighting VASA's shortcomings in cartoon generation, but this issue is unrelated to the contributions of NAR that we are discussing. Regarding the selected methods like Hallo, their slow speed is primarily due to the use of Stable Diffusion, which is inherently slow. Additionally, their intermediate representations are still image-level (albeit compressed). Furthermore, in terms of speed alone, previous GAN-based methods are actually much faster. While the authors have explained the differences between the modules in this paper and previous methods, the contributions remain relatively incremental. Furthermore, this paper does not provide much additional inspiration. Therefore, the current version of this paper may not yet be suitable for publication, and I may not raise my score.

---

> ### Author Response · Authors · 2024-11-24
> **Further clarification of our contribution**
>
> Dear reviewer 5zi1, thank you for your response and for acknowledging the amount of workload we have dedicated. In our response, we would like to politely clarify some potential misunderstandings and further elaborate on the contributions of our research.
>
> > VASA has already achieved real-time performance with satisfactory results, yet the authors insist on using NAR to address an issue that has already been resolved...VASA's shortcomings in cartoon generation, but this issue is unrelated to the contributions of NAR that we are discussing.
>
>
> We acknowledge that VASA is an outstanding work and fully understand your appreciation for the real-time performance it has achieved. However, some details need clarification. Firstly, VASA's "real-time performance" is achieved using a 4090 GPU with about 86 TFLOPS of computing power. In contrast, our performance was tested on a V100 16G, which has around 14 TFLOPS. This points to a significant difference in computational power between the two devices. Even though VASA is closed-source, we can estimate its speed on a V100, and it's might not real-time by those standards. The V100 16G's computing performance is similar to that of 4060/3060 GPUs, which more users currently have access to compared to 4090.
>
> Secondly, real-time speed might be adequate for scenarios where audio input speed has limits, such as video conferences. However, talking head tasks are used in other settings as well. For instance, in video production, users may have already recorded a lot of audio beforehand. In these cases, even with real-time generation, users have to wait for a long time. Thus, **real-time speed alone doesn't fully solve the problem of slow generation in such scenarios**. Therefore, developing faster generation strategies is still valuable and significant for research.
>
> Thirdly, we admit that NAR does not directly enhance generalization performance, such as driving cartoon content. However, VASA's speed boost largely comes from using compact 3D information to build the latent space. **This approach increases generation speed but sacrifices generalization**. In contrast, our acceleration strategy with NAR maintains generalization, highlighting the advantage of our solution.
>
> Beyond improving generation speed, the NAR method also **tackles the issue of error accumulation** in generated content. We consider this an important contribution that should not be neglected. We have verified this thoroughly from both quantitative and qualitative perspectives in our previous rebuttal and paper. While VASA has achieved good results, its AR nature means that it relies on historical results for generation. However, there is no guarantee that these historical results are completely precise and accurate. As a result, the AR methods it represents still carry the risk of error accumulation.
>
> > methods like Hallo, their slow speed is primarily due to the use of Stable Diffusion, which is inherently slow. Additionally, their intermediate representations are still image-level... previous GAN-based methods are actually much faster.
>
> We recognize that the speed of a method can be influenced by the number of parameters and the size or content of the latent state in different models. Thus, in Table 2 of the main paper, we have demonstrated that even when parameters and latent space size are nearly the same, NAR still surpasses other generation strategies in terms of both speed and quality.
>
> Moreover, diffusion-based methods usually deliver better generation results than GAN-based methods. While GANs are generally faster than AR-based diffusion methods, we noted in earlier versions of the paper that we have compared our speed with typical GAN-based methods, like MakeItTalk and Audio2head. Our findings demonstrated that our speed is comparable to, or even exceeds these methods, in terms of speed and quality.
>
> | Method | Inference time (s)  |
> |----------|----------|
> | MakeitTalk| 19.76     |
> | Audio2head    | 8.37     |
> | DAWN(ours)   | 8.31     |
>
> > While the authors have explained the differences between the modules in this paper and previous methods, the contributions remain relatively incremental. Furthermore, this paper does not provide much additional inspiration.
>
> Currently, most methods in the talking head field rely on AR, which could lead to error accumulation and slower generation speeds. These risks cannot be ignored. To tackle these issues, we developed an innovative NAR generation strategy that is fundamentally different from existing AR-based methods. We believe this strategy offers substantial novelty and extensive experiments demonstrate that NAR method effectively addresses these problems, thus providing valuable insights into the talking head field. As a new algorithm at the level of holistic generation strategies, we believe our NAR-based method holds significant academic and practical value, even if its current performance may not fully surpass the AR-based sota.

---

> ### Comment · Reviewer_5zi1 · 2024-11-24
>
> The author's response has made some factual errors: VASA does not rely on 3DMM but instead learns latent representations directly from the data. Additionally, the GAN-based methods compared in this comment are quite outdated, all from 2021 or earlier. Moreover, the author does not discuss a relevant work, Live Speech Portraits[1], which also addresses real-time talking head generation. The author's comments do not convince me. I believe that the current version of this paper is not suitable for publication.
>
> [1] Lu, Yuanxun, Jinxiang Chai, and Xun Cao. "Live speech portraits: real-time photorealistic talking-head animation." ACM Transactions on Graphics (ToG) 40.6 (2021): 1-17.

---

> > ### Author Response · Authors · 2024-11-24
> > **Thanks for your reply**
> >
> > Dear reviewer 5zi1, thank you for your careful reading. Indeed, our statement was not rigorous enough, but what we wanted to convey is that the VASA does rely on 3D information, and this increases the risk of a lack of generalization. Thank you for your suggestion, but our main contribution is not just to enhance speed. More importantly, we proposed a novel NAR generation framework and achieved performance comparable to, or even better than, existing AR-based methods.

---

### Official Review · Reviewer_Q1nh · 2024-10-24

**Soundness:** 3
**Presentation:** 3
**Contribution:** 3
**Rating:** 6
**Confidence:** 5

**Summary:**

The paper presents a new method for talking head generation called DAWN. This networks contains three main parts: a latent flow generator to drive a source image using latent flows, an "audio to latent flow" diffusion network and a pose and blink generation network. The network first predict latent flows from an audio sequence and pose and blink parameters. Those latent flows are then used by the latent flow generator to project the motion to RGB space using a source image containing the new identity. The pose and blink parameters are generated by the pose and blink generation network. DAWN outperforms the method used in the comparison on most metrics quantitatively and also qualitatively. The generated head pose and blink can be replaced by real one extracted from video for better control on the generation. Extensive ablation study is provided.

**Strengths:**

- The method works directly on the image by using its own latent flow generator. This is good because the network doesn't need to rely on other intermediate representations (e.g. 3dmm extracted from the video) which can already contain errors leading to bad generation.

- DAWN is able to generate pose and blink directly from the audio which is something not many methods do.

- By generating the head pose and blinking parameters outside the diffusion network and using them as condition instead the authors make their method more controllable.

- Qualitative results look good even if in low resolution.

- The authors provided video results for qualitative evaluation.

- The ablation study is extensive.

**Weaknesses:**

- The main issue I have with the paper is that the method used as baseline for the comparison are not state of the art: Wav2Lip, MakeItTalk and Audio2Head are from 2020-2021 while SadTalker and DiffusedHeads are from 2023. There are more recent method that could have been used for the quantitative comparison [1,2], there also exist diffusion based method that perform much better than DiffusedHeads [3]. Additionally several recent methods, for which the code is not available, could have been considered only for the qualitative results due to their impressive performances [4,5,6].

- On the same subject DiffusedHeads is known to perform badly on out of distribution data. It was trained on the MEAD dataset and tend to fail if the background is not green.

- The method is able to generate head pose but both datasets used for experiments exhibit limited head poses. It would have been better to use the VoxCeleb or CelebV-HQ datasets to evaluate the quality of the generated head poses.

- Most experiments are performed with a resolution of 128*128 when state of the art usually use 256X256 or higher (notably for HDTF that is often used at 512X512).

- In table 1 the LSEc and LSEd of wav2lips are not in bold/underlined when they should, they are often the best.

- For the video results provided the authors should make it clear whether the head pose was generated or taken from the ground truth. In some examples (e.g. driven_by_music.mp4) it look really impressive but I suspect it come from the ground truth since the results in the comparison video, while good, are not as impressive.

- It is still not clear how the non-autoregressive part is used during inference. Are the sequences generated only at a chosen size (e.g 200 for HDTF)? Or are several sequence generated then put back to back to make long videos (like the ones from the supplementary material). If it is the latter, since the method is non-autoregressive how do the authors avoid discontinuity between sequences with respect to the head pose?

- In table 4 the authors show that PBNET improve lips synchronization (LSEd and LSEc) by a lot but do not explain in details why this is happening. While it's true that the training become more difficult as a result there is still a loss focusing on lips so the lip motion should stay about as good in my opinion.

- I understand why the authors perform two training with different sequence length and the ablation shows that it improves results. However I don't see why they set X_src=X_1 for the first training instead of using a random starting frame like they do in the second training. Could the authors clarify their choice on this?

- The paper mention that a GAN loss is used inside PBNET but does not say how it is applied. Is there a discriminator? This part need to be clarified.

- The paper state that AR and SAR "leads to constrained performance and potential error accumulation, especially in long video sequences" but only cite one paper to support that claim. Most other methods mentioned in the paper or the one I proposed seem fine in that regard. The authors should develop this part with more examples.

[1]: Tan, Shuai, Bin Ji, and Ye Pan. "Style2talker: High-resolution talking head generation with emotion style and art style." Proceedings of the AAAI Conference on Artificial Intelligence. Vol. 38. No. 5. 2024.

[2] S. Wang et al., "StyleTalk++: A Unified Framework for Controlling the Speaking Styles of Talking Heads," in IEEE Transactions on Pattern Analysis and Machine Intelligence, vol. 46, no. 6, pp. 4331-4347, June 2024, doi: 10.1109/TPAMI.2024.3357808

[3]: Ma, Y., Zhang, S., Wang, J., Wang, X., Zhang, Y., Deng, Z.: Dreamtalk: When expressive talking head generation meets diffusion probabilistic models. arXiv preprint arXiv:2312.09767 (2023)

[4]: Xu, Sicheng, et al. "Vasa-1: Lifelike audio-driven talking faces generated in real time." arXiv preprint arXiv:2404.10667 (2024).

[5]: Zhang, Bingyuan, et al. "Emotalker: Emotionally editable talking face generation via diffusion model." ICASSP 2024-2024 IEEE International Conference on Acoustics, Speech and Signal Processing (ICASSP). IEEE, 2024

[6]: Tian, Linrui, et al. "Emo: Emote portrait alive-generating expressive portrait videos with audio2video diffusion model under weak conditions." arXiv preprint arXiv:2402.17485 (2024).

**Questions:**

- The latent flow generator is effectively a video-driven talking head generation method. It would have been interesting to see how it perform against similar methods, e.g. [7].

- Notation issue: in eq .6 the input are x_src,y_1N and p_1N. in the conditioning paragraph and in the figure it is stated that instead the inputs are in fact Z_src,a_1N and p_1N. Also p_1N is not defined when it's used in eq.6. Both of these things need to be corrected.

[7]: Y. Wang, D. Yang, F. Bremond and A. Dantcheva, "LIA: Latent Image Animator," in IEEE Transactions on Pattern Analysis and Machine Intelligence, doi: 10.1109/TPAMI.2024.3449075.

---

> ### Author Response · Authors · 2024-11-17
> **Author Response to Official Review by Reviewer Q1nh (1/5)**
>
> Dear reviewer Q1nh, thank you for reviewing our work and providing numerous valuable suggestions. We believe these suggestions will greatly help us in enhancing the quality of our paper. Below are our responses to your questions, which we hope will address your concerns. Additionally, we have uploaded new supplementary videos to the [anonymous repository](https://anonymous.4open.science/r/DAWN-BC9F/README.md). Since the files are large, for a better viewing experience, we recommend downloading the supplementary materials first.
>
> > Q1：The main issue I have with the paper is that the method used as baseline for the comparison are not state of the art.
>
>
> Thank you for your suggestion. Considering that these methods either use the image cropping technique [1,10,11] to centralize the portrait or employ Stable Diffusion pre-trained models [2,3], we initially chose not to compare with them for the sake of rigor, as this could affect the fairness of the comparison. However, we agree that comparing with the latest SOTA methods is essential, so we have added a comprehensive comparison experiment with **Dreamtalk (2023) [1]**, **Hallo (2024.6) [2]**, and **Echomimic (2024.7) [3]**.
>
> It is important to note that EchoMimic and DreamTalk are only applicable to face-aligned scenarios, whereas other methods do not require pre-cropping. Additionally, Hallo and EchoMimic are built upon pre-trained Stable Diffusion models, inheriting substantial visual generation capabilities. Despite this, our method still achieves comparable or even better performance on the HDTF dataset.
>
> |          | FID    | FVD16 |FVD32 |LSE-C | LSE-D | CSIM     | BAS      | Blink/s  |
> |----------|----------|----------|----------|----------|----------|----------|----------|----------|
> | GT        | -        | -        | -        | 7.95     | 7.33     | 1       | 0.267     | 0.75     |
> | DreamTalk | 58.8     | 406.58     | 516.21     | 6.48     | 8.43     | 0.641     | 0.3110     | 0.032     |
> | Hallo     | 14.2     | **57.47**     | 100.99     | **7.16**     | 8.01     | 0.7095     | 0.3010     | 0.254     |
> | Echomimic | 32.8     | 139.00     | 178.16     | 6.69     | 8.27     | 0.7314     | 0.3177     | 0.1212     |
> | DAWN(ours)    | **9.60**    | 60.34     | **95.64**    | 6.71     | **7.94**    | **0.790**     | **0.281**     | **0.86**     |
>
> > Q2: On the same subject DiffusedHeads is known to perform badly on out of distribution data. It was trained on the MEAD dataset and tend to fail if the background is not green.
>
> We agree that using only DiffusedHeads to illustrate the error accumulation issue in autoregressive (AR) diffusion models may lack persuasive power. Therefore, we have additionally selected **Hallo**[2] for comparison. Most AR or semi-autoregressive (SAR) DM methods have a similar structure to Hallo, which is influential in the community and can represent the typical performance of them. The example of error accumulation can be found in the re-uploaded supplement material ([*Compare_with_Hallo.mp4*](https://anonymous.4open.science/r/DAWN-BC9F/Compare_with_Hallo.mp4)). As shown, the image collapse issue caused by error accumulation is a common problem in AR-based models, not just specific to DiffusedHeads.
>
> > Q3: The method is able to generate head pose but both datasets used for experiments exhibit limited head poses. It would have been better to use the VoxCeleb or CelebV-HQ datasets to evaluate the quality of the generated head poses.
>
> We agree that using a dataset with richer head movements can better showcase the performance of PBNet, so we conducted some qualitative tests on **CelebV-HQ**, as shown in [*"CelebV_HQ_SR.mp4"*](https://anonymous.4open.science/r/DAWN-BC9F/CelebV_HQ_SR.mp4) from re-uploaded supplement material. Our approach is indeed capable of generating lively head movements. However, due to the more complex backgrounds in CelebV-HQ, and the additional task of including hand and upper body movements, using a checkpoint trained on the HDTF dataset presents some challenges. We plan to address these issues in future work.
>
>
>
> \
> [1] Ma, Yifeng, et al. "Dreamtalk: When expressive talking head generation meets diffusion probabilistic models." arXiv preprint arXiv:2312.09767 (2023).
>
> [2] Xu, Mingwang, et al. "Hallo: Hierarchical audio-driven visual synthesis for portrait image animation." arXiv preprint arXiv:2406.08801 (2024).
>
> [3] Chen, Zhiyuan, et al. "Echomimic: Lifelike audio-driven portrait animations through editable landmark conditions." arXiv preprint arXiv:2407.08136 (2024).
>
> [10] Ma, Yifeng et al. “StyleTalk: One-shot Talking Head Generation with Controllable Speaking Styles.” AAAI Conference on Artificial Intelligence (2023).
>
> [11] Wang, Suzhe et al. “StyleTalk++: A Unified Framework for Controlling the Speaking Styles of Talking Heads.” IEEE Transactions on Pattern Analysis and Machine Intelligence 46 (2024): 4331-4347.

---

> ### Author Response · Authors · 2024-11-17
> **Author Response to Official Review by Reviewer Q1nh (2/5)**
>
> > Q4： Most experiments are performed with a resolution of 128*128 when state of the art usually use 256X256 or higher (notably for HDTF that is often used at 512X512).
>
> Thank you for your suggestion. We originally provide the qualitative study of **256×256** in the appendix, Figure 4. We agree that testing the model's performance at higher resolutions is necessary, so we have added the overall metrics for our method at **256×256** resolution on the HDTF dataset. From the results, it is clear that our method still maintains strong competitiveness compared to the latest talking head models. When the resolution is higher, more computational resources are required. Therefore, for **512×512** resolution, we tried using some super-resolution techniques, such as **Codeformer [4]**, and achieved promising results. We have included these results in the re-uploaded supplement material [*"CelebV_HQ_SR.mp4"*](https://anonymous.4open.science/r/DAWN-BC9F/CelebV_HQ_SR.mp4), which has been referred in Q3.
>
> |          | FID      | FVD16 |FVD32 |LSE-C | LSE-D | CSIM     | BAS      | Blink/s  |
> |----------|----------|----------|----------|----------|----------|----------|----------|----------|
> | GT        | -        | -        | -        | 7.95     | 7.33     | 1       | 0.267     | 0.75     |
> | DAWN 128    | 9.60    | 60.34     | 95.64    | 6.71     | 7.94    | 0.790     | 0.281     | 0.86     |
> | DAWN 256    | 11.80    | 68.07     | 105.20    | 7.20     | 7.80    | 0.791     | 0.278     | 0.73     |
>
> > Q5：In table 1 the LSEc and LSEd of wav2lips are not in bold/underlined when they should, they are often the best.
>
> Although the comparison with Wav2Lip is not entirely fair, as Wav2Lip only drives the lips, we believe that Wav2Lip has extremely accurate lip movement generation capabilities. Therefore, we use it as a reference rather than a rival for lip-sync performance in the comparison to better reflect the performance of our method. To avoid misunderstanding, we have clarified it in the title of Table 1 and marked it with an asterisk (*). We will explain wavlip’s detailed settings more explicitly in the comparative experiments section.
>
> > Q6: For the video results provided the authors should make it clear whether the head pose was generated or taken from the ground truth. In some examples (e.g. driven_by_music.mp4) it look really impressive but I suspect it come from the ground truth since the results in the comparison video, while good, are not as impressive.
>
> Thank you for your recognition of our results. We promise that, except for the "[*Cross-identity_reenactment.mp4*](https://anonymous.4open.science/r/DAWN-BC9F/Cross-identity_reenactment.mp4)", where we used ground-truth pose to demonstrate the pose control effect, the head poses of all other videos are generated by our model. We will re-upload the supplementary materials and clarify this point more explicitly. We believe that in "[*Driven_by_music.mp4*](https://anonymous.4open.science/r/DAWN-BC9F/Driven_by_music.mp4)", the driving audio is a song, which inherently has more emotion and rhythmic variation, leading to richer head movements. In contrast, a more neutral speech might result in less head movement.
>
>
> > Q7: It is still not clear how the non-autoregressive part is used during inference.
>
> Our method generates dynamic-length video sequences in a single generation during the inference, and the video length depends on the length of input audio sequence. This provides an inherent advantage in terms of video continuity, which is one of the key motivations for proposing our non-autoregressive inference architecture. In *"[Extrapolation_evaluation.mp4](https://anonymous.4open.science/r/DAWN-BC9F/Extrapolation_evaluation.mp4)"*, each video was generated in a single pass. For example, given an 8-second audio input, our model generates an 8-second video in a single pass. Similarly, for a 16-second or longer audio input, the model produces a correspondingly longer video sequence in a single generation. We will clarify this in the paper to prevent potential misunderstanding for readers.
> \
> \
> \
> [4] Zhou, Shangchen, et al. "Towards robust blind face restoration with codebook lookup transformer." Advances in Neural Information Processing Systems 35 (2022): 30599-30611.

---

> ### Author Response · Authors · 2024-11-17
> **Author Response to Official Review by Reviewer Q1nh (3/5)**
>
> > Q8: In table 4 the authors show that PBNET improve lips synchronization (LSEd and LSEc) by a lot but do not explain in details why this is happening. While it's true that the training become more difficult as a result there is still a loss focusing on lips so the lip motion should stay about as good in my opinion.
>
> Yes, we have designed an exclusive loss function for the lip region. However, the loss functions in the diffusion model (DM) are divided into two components: $L_1$, which is the overall denoising loss, and $L_2$, which focuses on denoising the lip region. If the DM simultaneously generates pose, blink, and lip movements, the task complexity for generation increases significantly. This causes $L_1$ to dilute the influence of $L_2$ on the model, thereby reducing the effectiveness of $L_2$. Consequently, this results in a worse LSE-C and LSE-D.
>
> > Q9: I don't see why they set X_src=X_1 for the first training instead of using a random starting frame like they do in the second training. Could the authors clarify their choice on this?
>
> This is because Stage 1 focuses on simple and basic lip movements. Using $x_{\text{src}} = x_{1}​$ in Stage 1 ensures that $x_{\text{src}}​$ corresponds as closely as possible to the first frame of the audio information. This is equivalent to providing the model with a ground truth initial frame, which reduces the difficulty of training and helps the model learn the basic lip movement modeling ability, ensuring the convergence of Stage 2.
>
> However, in the inference process, the input $x_{\text{src}}$ may represent a person in any arbitrary pose. If we always provide the first frame image $x_1$ as $x_{\text{src}}$, the model would become overly reliant on the pose and blink state of the character in that initial frame. To prevent this dependency and to improve control over pose and blink, we introduce random $x_{\text{src}}$ in Stage 2. This allows the model to handle more dynamic and diverse input while improving its pose and blink control capabilities. We hope this explanation addresses your concern.
>
> > Q10: The paper mention that a GAN loss is used inside PBNET but does not say how it is applied. Is there a discriminator? This part need to be clarified.
>
> Yes, we applied a discriminator to guide the training of the model. The discriminator is based on PatchGAN [5]. PatchGAN outputs the probability of each patch in the pose/blink sequence originating from real actions. The discriminator is guided by the BCE loss. We will elaborate on this part in the appendix to minimize potential confusion.
> \
> \
> \
> [5] Isola, Phillip, et al. "Image-to-image translation with conditional adversarial networks." Proceedings of the IEEE conference on computer vision and pattern recognition. 2017.

---

> > ### Comment · Reviewer_Q1nh · 2024-11-19
> > **questions about rebuttal**
> >
> > Q9. Couldn't this be solved with a weighting strategy ?

---

> ### Author Response · Authors · 2024-11-17
> **Author Response to Official Review by Reviewer Q1nh (4/5)**
>
> > Q11: The paper state that AR and SAR "leads to constrained performance and potential error accumulation, especially in long video sequences" but only cite one paper to support that claim. Most other methods mentioned in the paper or the one I proposed seem fine in that regard. The authors should develop this part with more examples.
>
> Thank you for your suggestions. The issue of **error accumulation** in AR/SAR methods has been explicitly mentioned in the following works: DiffusedHead[6], Hallo[2], and EMO[7]. We agree that supplementing additional materials to explain the error accumulation phenomenon in previous studies is necessary. Below, we will discuss the error accumulation phenomenon in AR/SAR methods from both **qualitative** and **quantitative** perspectives.
>
> We selected Hallo for comparison, as it represents the basic structure of most AR or SAR diffusion model structures and has significant influence in the community. From the qualitative perspective, we have provided the example in  Q2.
>
> In addition, from the quantitative perspective, we design a metric to quantify the severity of error accumulation inspired by the work [9] in sequential generation: **Degradation Rate (DR)**, defined as: $\text{DR} = \frac{\text{FID}\_{\text{ed}}}{\text{FID}\_{\text{st}}} -1$.
>
> The motivation for proposing this metric is that when error accumulation occurs, the quality of the generated data at the end of the sequence significantly deteriorates compared to the beginning. DR is related to the ratio between the Fréchet Inception Distance (FID) of the last $n$ frames $\text{FID}\_{\text{ed}}$ and the first $ n$ frames ​$\text{FID}\_{\text{st}}$. A larger DR indicates more severe error accumulation. In our experiments, we set $n$ to 25 frames (1s) and 50 frames (2s), denoted as $\text{DR}\_{25}$ and $\text{DR}\_{50}$, respectively.
>
> **$\text{DR}\_{25}$:**
>
> | Inference length | DAWN  | Hallo  |
> |----------|----------|----------|
> | 200    | 0.253     | 0.214     |
> | 400    | 0.208     | 0.279     |
> | 600    | 0.152     | 0.422     |
>
> \
> **$\text{DR}\_{50}$:**
>
> | Inference length | DAWN  | Hallo  |
> |----------|----------|----------|
> | 200    | 0.043     | 0.094     |
> | 400    | 0.164     | 0.161     |
> | 600    | 0.144     | 0.332     |
>
> In the extrapolation evaluation results, it can be observed that the DR metric increases progressively with the inference length for Hallo. However, the DR metric of our method **stabilizes within a certain range**. This indicates that Hallo experiences significant error accumulation as the inference length increases, whereas our performance remains stable.
>
> Additionally, we have added a comparison of generation strategies in Table 2. It can be observed that the SAR inference strategy shows significant error accumulation, while our method exhibits minimal error accumulation, thanks to the NAR generation. Although TTR can also avoid error accumulation, it does not perform as well as the NAR approach in terms of generation speed and quality. These experiments further reinforce the motivation behind our proposed NAR approach. We will include the supplementary experiment related to DR in Tables 2 and 3 of the paper.
>
> | Method   | $\text{DR}\_{25}$  | $\text{DR}\_{50}$  |
> |----------|----------|----------|
> | SAR    | 0.307     | 0.201    |
> | TTR    | -0.028   | -0.005  |
> | Ours   | 0.044     | 0.031    |
>
> \
> \
> \
> [6] Stypułkowski, Michał, et al. "Diffused heads: Diffusion models beat gans on talking-face generation." Proceedings of the IEEE/CVF Winter Conference on Applications of Computer Vision. 2024.
>
> [7] Tian, Linrui, et al. "Emo: Emote portrait alive-generating expressive portrait videos with audio2video diffusion model under weak conditions." arXiv preprint arXiv:2402.17485 (2024).
>
> [9] Bian, Xiaohang, et al. "Handwritten mathematical expression recognition via attention aggregation based bi-directional mutual learning." Proceedings of the AAAI conference on artificial intelligence. Vol. 36. No. 1. 2022.

---

> > ### Comment · Reviewer_Q1nh · 2024-11-19
> > **Questions about rebuttal**
> >
> > Q11. Same comment as for Q2. "the authors should specify that this apply to images based diffusion model as 3D based diffusion model such as the one used by Dreamtalk do not seem to suffer from it."

---

> ### Author Response · Authors · 2024-11-17
> **Author Response to Official Review by Reviewer Q1nh (5/5)**
>
> > Q12: The latent flow generator is effectively a video-driven talking head generation method. It would have been interesting to see how it perform against similar methods, e.g. LIA.
>
> Thank you for your suggestion. We have compared the performance of LIA[8] and LFG on the HDTF dataset, using GT videos to drive the first-frame image. To eliminate the influence of pretraining on HDTF, both LIA and LFG use checkpoints pretrained on the VoxCeleb dataset. The results for LFG-finetune were obtained by fine-tuning on HDTF. The experimental results are as follows:
>
> | Method  |   FID      |FVD16|FVD32 |LSE-C | LSE-D |
> |----------|----------|----------|----------|----------|----------|
> | LIA    |26.64     | 95.75   | 178.62   | 6.69     | 8.12     |
> | LFG    |12.69    |55.24      | 105.82   | 6.72     | 8.32     |
> | LFG-finetune    |7.23    |24.80      | 38.00   | 7.36     | 7.73     |
>
> In our method, LFG is crucial for extracting motion information and providing supervisory signals to the diffusion model. We emphasize LFG's video reconstruction capabilities, as this module should ensure near-lossless video reconstruction to avoid limiting the overall model performance. We believe that LIA may indeed outperform LFG in certain scenarios (e.g., cross-identity reenactment and handling occlusion relationships). Therefore, we plan to leverage the advantages of LIA to improve LFG in future work, aiming to develop a more effective talking head generation method.
>
> > Q13: Notation issue: in eq .6 the input are x_src,y_1N and p_1N. in the conditioning paragraph and in the figure it is stated that instead the inputs are in fact Z_src,a_1N and p_1N. Also p_1N is not defined when it's used in eq.6. Both of these things need to be corrected.
>
> Thank you for carefully reviewing our paper. Based on your suggestions, we will add the relevant definitions and clarifications in the paper to ensure its rigor.
> \
> \
> \
> [8] Wang, Yaohui et al. “LIA: Latent Image Animator.” IEEE Transactions on Pattern Analysis and Machine Intelligence 46 (2024): 10829-10844.

---

> ### Comment · Reviewer_Q1nh · 2024-11-19
> **Questions about rebuttal**
>
> I thanks the authors for their extensive rebuttal. I still have some more concerns/question that I would like to have answered.
>
> Q1. The authors mention that "EchoMimic and DreamTalk are only applicable to face-aligned scenarios" this imply that DAWN can work in non face aligned scenarios. However, in the qualitative results (figure and supplementary materials), there are only example in face aligned scenario. It would have been good to have non face algined example to support that claim and show the superiority of DAWN over EchoMimic and DreamTalk.
>
> Q2. I thank  the authors for providing more example of error accumulation with AR diffusion model. However the authors should specify that this apply to images based diffusion model as 3D based diffusion model such as the one used by Dreamtalk do not seem to suffer from it.

---

> ### Author Response · Authors · 2024-11-20
> **Thanks for your reply**
>
> Dear reviewer Q1nh, thank you for taking the time and effort to read our rebuttal. We hope that our further responses can address your concerns.
>
> **Q1:**
> Thank you for your suggestions. To better illustrate the advantages of our method in non-face-aligned scenarios, we have added supplementary materials to enhance the comparison between DAWN, DreamTalk, and Echomimic ([*Comparison_of_unaligned_face_scenarios.mp4*](https://anonymous.4open.science/r/DAWN-BC9F/Comparison_of_unaligned_face_scenarios.mp4)). We have prepared three examples. The face is not aligned and the head area occupies only about nearly a quarter of the frame. We disabled image cropping during testing. We found that, because DreamTalk only models the 3D representation of the head, its robustness is insufficient to drive images that include regions below the shoulders. In Echomimic, the overall frame tends to be static; although there are some minor head movements, there is almost no lip movement. This indicates that the Echomimic method heavily depends on the pre-processing of $x_{src}$. In contrast, our method generates vivid head movements and accurate lip driving in each example. It is noteworthy that even in the third example, where a large part of the torso is involved, our method still performs very well.
>
> **Q2:**
> Thank you for pointing this out. We acknowledge that the degradation rate based on FID may not detect the error accumulation phenomenon in autoregressive methods that generate 3D representations, such as DreamTalk. The error accumulation in these methods is more likely related to the accuracy and vividness of facial dynamics, rather than quality degradation in the image space. We promise to add corresponding notes in the paper to clarify this point.
>
> **Q9:**
>
> Thank you for your suggestion. We presume that you might be suggesting increasing the weight of the lip loss during the first stage as a way to enhance the model's driving effect for lip movement, instead of using $x_1$ as $x_{src}$. Increasing the weight of lip loss may result in better lip motions. However, this does not further reduce the training difficulty of the first stage. This is because the model does not actually receive additional conditioning this way. In contrast, if we use $x_1$ as $x_{src}$, it is equivalent to providing the ground truth information of the first frame, and the model only needs to generate other frames based on the first frame. This approach reduces the requirement for the model to produce large motion fields. Furthermore, in stage 1, we train with only a 20-frame video clip, within which there typically aren’t varied head pose changes. Therefore, the model only needs to focus on synthesizing accurate lip movements.
>
> If a random $x_{src}$ is introduced in the first stage, since $x_{src}$ can be presented in any pose, the model needs to additionally consider how to warp $x_{src}$ to match the pose of the ground truth initial frame. This would force the model to learn how to accurately perform arbitrary head pose warping, which in turn prevents the model from focusing on learning the driving of lip movements.
>
>
> The lip loss in this context comes from Diffused Heads [1], and we employ a weight of w = 0.2 during training. Diffused Heads demonstrated that w = 0.2 yields the best lip movements and highly realistic quality. Additionally, it also reported that a higher weight will degrade the overall video quality. Thus, we consider the weight of the lip loss in our experiment to be nearly optimal.
> \
> \
> \
> Additionally, we have supplemented the analysis of the weighting strategy for **Q8**:
>
> Thank you for your suggestion. Increasing the weight of lip loss may result in higher LSE-C/LSE-D, but we want to emphasize that removing PBNet leads to significant changes in the learning objectives of the diffusion model. When PBNet is present, the diffusion model is not responsible for generating pose/blink, but rather for explicitly driving them. This task is much easier than generating them directly, thus allowing the model to focus more on generating better lip movement. Therefore, although PBNet does not directly generate lip movements, its removal forces the model to balance the priorities between pose, blink and lip motion generation, ultimately leading to a decline in lip motion performance.
>
> Since Diffused Heads use DM to simultaneously generate lip, pose, and blink motions, we believe this is more closely aligned with the "w/o PBNet" setting. Therefore, we believe the weight of w = 0.2 for lip loss in experiments has fully exploited the model's potential for generating lip movements. Unlike Diffused Heads, in our method, the DM does not generate pose/blink from speech, which might improve lip driving by adjusting weights. For fairness in ablation studies, we kept the lip loss weight consistent across experiments.
> \
> \
> \
> [1] Stypulkowski, Michal et al. “Diffused Heads: Diffusion Models Beat GANs on Talking-Face Generation.” 2024 WACV

---

> ### Author Response · Authors · 2024-11-22
> **We hope that our response addresses your concern**
>
> Dear Reviewer Q1nh,
>
> We earnestly thank you for taking the time to read and respond to our rebuttal. We have carefully provided further explanation regarding your concern. We are eager to know if our response has addressed your concern. As the end of the rebuttal phase is approaching, we look forward to hearing from you for any further clarification that you might require.
>
> Best,
>
> Submission 6731 authors

---

> ### Author Response · Authors · 2024-11-27
> **Reminder:  deadline of revision for paper is approaching**
>
> Dear Reviewer Q1nh,
>
> We sincerely appreciate your support in improving the quality of our paper. In accordance with your recommendations, we have made the corresponding modifications to the article (Main Paper line 349, Appendix lines 801 and 807). As the deadline for submitting the paper PDF is approaching, we would be grateful if you could share any further concerns or suggestions you might have, and we will be more than happy to address them promptly. We are looking forward to your feedback and response. Finally, we extend our heartfelt thanks once again for taking the time to review our response.
>
> Best,
>
> Submission 6731 authors

---

> ### Comment · Reviewer_Q1nh · 2024-11-29
> **Rating change**
>
> After carefully reviewing the authors rebuttals and revised version of the paper, I raise my rating to 6. The new experiments provided by the authors make the paper stronger.

---

> > ### Author Response · Authors · 2024-12-01
> > **Thanks for your reply**
> >
> > Dear Reviewer Q1nh,
> >
> > Thanks a lot for your engagement and positive comments! We would like to once again express our gratitude for the time and effort you have devoted to reviewing our paper, as well as for providing numerous valuable suggestions that have helped us improve the quality of the article.
> >
> > Best,
> >
> > Submission 6731 authors

---

### Official Review · Reviewer_fcWS · 2024-10-25

**Soundness:** 2
**Presentation:** 3
**Contribution:** 2
**Rating:** 5
**Confidence:** 5

**Summary:**

This paper proposes a diffusion-based non-autoregressive talking head synthesis method that operates with input from only a source portrait and an audio sequence. The method is performed in a two-stage manner: a Latent Flow Generator (LFG) serves as the video reenactment model, and a diffusion model together with PBNet serve as the motion generation models. To achieve non-autoregressive synthesis (NAR), PBNet generates the entire blink and pose sequence at once. The authors also propose a two-stage curriculum learning strategy to enhance the efficiency and performance of the diffusion model's learning process.

**Strengths:**

1. The paper is well written and easy to read.
2. The proposed method can performed nearly in real time on a single V100 16G GPU.

**Weaknesses:**

1. I believe the proposed method lacks novelty. The overall architecture and the two-stage approach, which involves training a reenactment model first and then generating motion latents, have already been proposed in many existing works, such as TH-PAD[1], GAIA[2], VASA-1[3], AniTalker[4], and so on.. Although the authors attempted to innovate from a non-autoregressive perspective, I believe that their discussion and argumentation regarding the advantages of NAR are not sufficiently robust, and I will state my reasons below.
2. I don't think the proposed NAR method and experiments conducted demonstrate the authors' claim:
a) Generating only blinks and poses through PBNet is not sufficient; the expression-related motion is still generated from the diffusion branch with audio input as a condition.
b) The authors claim that there is error accumulation in long video sequences of existing AR and SAR models, but no comparative experiments have been conducted between the proposed NAR method and existing methods to prove this claim.
c) Given that the authors consider NAR as the main contribution of this paper, the paper does not adequately discuss the advantages of NAR over AR and SAR, nor does it provide reasonable comparative metrics to substantiate these claims. Merely comparing general metrics such as image quality, lip sync, identity, and motion matching is not sufficient to demonstrate that the performance improvements are due to the adoption of NAR.
3. Methods like Anitalker[4], TH-PAD[1], and VASA-1[3], etc., have similar overall architectures with the proposed one, but the authors did not illustrate the differences between the proposed method and these methods.
4. Recently, many diffusion-based talking head synthesis methods have been proposed, such as AniPortrait[5], Anitalker[4], Hallo[6], EchoMimic[7], FollowYourEmoji[8], and so on. Most of them operate in an auto-regressive manner. The authors did not compare the overall performance with these methods; therefore, there is no adequate evidence to prove the performance superiority of the proposed method.

Generally, I think the authors should conduct more experiments and propose more reasonable metrics to prove the effectiveness of the proposed method. And considering the main contribution is the non-autoregressive approach, the authors need to adequately discuss its advantages over AR and SAR. Based on the above reasons, I have doubts that this submission meets the bar for publication.

If the authors could discuss the advantages of NAR more thoroughly and supplement with sufficient experiments to prove that the proposed method has better effects compared to existing diffusion methods, I would raise my rating.

[1] Yu, Zhentao, et al. "Talking head generation with probabilistic audio-to-visual diffusion priors." Proceedings of the IEEE/CVF International Conference on Computer Vision. 2023.
[2] He, Tianyu, et al. "Gaia: Zero-shot talking avatar generation." arXiv preprint arXiv:2311.15230 (2023).
[3] Xu, Sicheng, et al. "Vasa-1: Lifelike audio-driven talking faces generated in real time." arXiv preprint arXiv:2404.10667 (2024).
[4] Liu, Tao, et al. "AniTalker: Animate Vivid and Diverse Talking Faces through Identity-Decoupled Facial Motion Encoding." arXiv preprint arXiv:2405.03121 (2024).
[5] Wei, Huawei, Zejun Yang, and Zhisheng Wang. "Aniportrait: Audio-driven synthesis of photorealistic portrait animation." arXiv preprint arXiv:2403.17694 (2024).
[6] Xu, Mingwang, et al. "Hallo: Hierarchical Audio-Driven Visual Synthesis for Portrait Image Animation." arXiv preprint arXiv:2406.08801 (2024).
[7] Chen, Zhiyuan, et al. "Echomimic: Lifelike audio-driven portrait animations through editable landmark conditions." arXiv preprint arXiv:2407.08136 (2024).
[8] Ma, Yue, et al. "Follow-Your-Emoji: Fine-Controllable and Expressive Freestyle Portrait Animation." arXiv preprint arXiv:2406.01900 (2024).

**Questions:**

1. I have a question regarding the ablation study on the TCL strategy: when the model was trained only with stage 1 and only with stage 2, did you maintain the same number of training epochs and steps as in the proposed method? I am curious to know whether the effectiveness of the two-stage curriculum learning is due to the training time or the strategy itself.

New after rebuttal: I raised the soundness score from 1 to 2 and the rating from 3 to 5.

---

> ### Author Response · Authors · 2024-11-17
> **Author Response to Official Review by Reviewer fcWS (1/3)**
>
> Dear reviewer fcWS, we appreciate your thorough review and valuable suggestions, which are very helpful in improving the quality of our paper. Although the concept of training a reenactment model and using a diffusion model to generate motion latent has been somewhat reflected in many works, our motivation, training strategies, and implementation methods differ significantly from these works. We hope that our following clarifications can address your concerns. Additionally, we have uploaded new supplementary videos to the [anonymous repository](https://anonymous.4open.science/r/DAWN-BC9F/README.md). Since the files are large, for a better viewing experience, we recommend downloading the supplementary materials first.
>
> > Q1： Generating only blinks and poses through PBNet is not sufficient; the expression-related motion is still generated from the diffusion branch with audio input as a condition.
>
>
> Both our PBNet and A2V-FDM can operate in a fully NAR manner. The main reasons for this design are:
> 1. Head pose and blink can be converted into low-dimensional representations. However, these actions often have longer durations, requiring training on long sequences to generate vivid motion. Using a lightweight PBNet to exclusively train to generate the pose and blink can significantly reduce training costs and difficulty.
> 2. Facial expressions are more complex, so we opt to fully leverage the generative capabilities of the diffusion model to produce realistic facial dynamics to ensure the model's effectiveness.
>
>
> > Q2： The authors claim that there is error accumulation in long video sequences of existing AR and SAR models, but no comparative experiments have been conducted between the proposed NAR method and existing methods to prove this claim.&#x20;
>
>
> To better demonstrate the advantages of the NAR approach compared to existing AR and SAR methods, we incorporated several open-source DM-based methods into our experiments, including **DreamTalk (2023)** [1], **Hallo (2024.6)** [2], and **EchoMimic (2024.7)** [3]. The results are as follows:
>
> |          | FID      | FVD16 |FVD32 |LSE-C | LSE-D | CSIM     | BAS      | Blink/s  |
> |----------|----------|----------|----------|----------|----------|----------|----------|----------|
> | GT        | -        | -        | -        | 7.95     | 7.33     | 1       | 0.267     | 0.75     |
> | DreamTalk | 58.8     | 406.58     | 516.21     | 6.48     | 8.43     | 0.641     | 0.3110     | 0.032     |
> | Hallo     | 14.2     | **57.47**     | 100.99     | **7.16**     | 8.01     | 0.7095     | 0.3010     | 0.254     |
> | Echomimic | 32.8     | 139.00     | 178.16     | 6.69     | 8.27     | 0.7314     | 0.3177     | 0.1212     |
> | DAWN(ours)    | **9.60**    | 60.34     | **95.64**    | 6.71     | **7.94**    | **0.790**     | **0.281**     | **0.86**     |
>
>
> It is important to note that EchoMimic and DreamTalk are **only applicable to face-aligned scenarios**, whereas other methods do not require pre-cropping. Additionally, Hallo and EchoMimic  **are built upon pre-trained Stable Diffusion models**, inheriting substantial visual generation capabilities. Despite this, our method still achieves comparable performance to DreamTalk, Hallo, and EchoMimic on the HDTF dataset.
> \
> \
> [1] Ma, Yifeng, et al. "Dreamtalk: When expressive talking head generation meets diffusion probabilistic models." arXiv preprint arXiv:2312.09767 (2023).
>
> [2] Xu, Mingwang, et al. "Hallo: Hierarchical audio-driven visual synthesis for portrait image animation." arXiv preprint arXiv:2406.08801 (2024).
>
> [3] Chen, Zhiyuan, et al. "Echomimic: Lifelike audio-driven portrait animations through editable landmark conditions." arXiv preprint arXiv:2407.08136 (2024).

---

> > ### Comment · Reviewer_fcWS · 2024-11-20
> > **Question about rebuttal**
> >
> > Thank the authors for their rebuttal. Regarding the Q1, I still have concerns.
> > The authors claim PBNet was proposed to compensate for the limitations of extrapolation in NAR strategies, in other words, PBNet generates the entire sequence of blinks and head poses at once, which achieves long video generation in a NAR manner. In my view, from this perspective, expression-related movements, similar to head poses, have a weak correlation with audio input and also establish temporal connections with how movements were made over a certain period in the past, unlike lip movements which have a strong mapping relationship with audio and can be generated frame by frame. So expression-related motion should also be generated together with blinks and head pose in advance, ensuring the completeness of authors' claim.

---

> ### Author Response · Authors · 2024-11-17
> **Author Response to Official Review by Reviewer fcWS (2/3)**
>
> > Q3： Merely comparing general metrics such as image quality, lip sync, identity, and motion matching is not sufficient to demonstrate that the performance improvements are due to the adoption of NAR.
>
>
> Thank you for your suggestion. We agree that metrics such as image quality and lip sync may not directly reflect the occurrence of error accumulation. Therefore, inspired by the work[5] in sequential generation, we design a metric to quantify the severity of error accumulation: **Degradation Rate (DR)**, defined as:
> $\text{DR} = \frac{\text{FID}_{\text{ed}}}{\text{FID}\_{\text{st}}} -1$.
>
> The motivation for proposing this metric is that when error accumulation occurs, the quality of the generated data at the end of the sequence significantly deteriorates compared to the beginning. DR is related to the ratio between the Fréchet Inception Distance (FID) of the last $n​$ frames $\text{FID}\_{\text{ed}}​$ and the first $ n​$ frames ​$\text{FID}\_{\text{st}}​$. A larger DR indicates more severe error accumulation. In our experiments, we set $n​$ to 25 frames (1s) and 50 frames (2s), denoted as $\text{DR}\_{25}​$ and $\text{DR}\_{50}​$, respectively. Additionally, we selected Hallo[2] for comparison, as it represents most autoregressive or semi-autoregressive diffusion model structures and has significant influence in the community.
>
>
> To demonstrate the error accumulation issue in AR methods, we selected Hallo[2] as a representative autoregressive diffusion model and expanded the experiments in Table 3 using the DR metric. In the extrapolation evaluation results, it can be observed that the DR index increases progressively with the inference length for Hallo. However, the DR metric of our method **stabilizes within a certain range**. This indicates that Hallo experiences significant error accumulation as the inference length increases, whereas our performance remains stable.
>
> **$\text{DR}\_{25}​$：**
> | Inference length | DAWN  | Hallo  |
> |----------|----------|----------|
> | 200    | 0.253     | 0.214     |
> | 400    | 0.208     | 0.279     |
> | 600    | 0.152     | 0.422     |
>
> **$\text{DR}\_{50}​$：**
>
> | Inference length | DAWN  | Hallo  |
> |----------|----------|----------|
> | 200    | 0.043     | 0.094     |
> | 400    | 0.164     | 0.161     |
> | 600    | 0.144     | 0.332     |
>
> For qualitative analysis, we provided examples to illustrate the error accumulation phenomenon in the re-uploaded supplement material, "[Compare_with_Hallo.mp4](https://anonymous.4open.science/r/DAWN-BC9F/Compare_with_Hallo.mp4)".
>
>
> Additionally, we further extended Table 2 using the DR metric to provide a more comprehensive discussion of error accumulation:
>
> | Method   | $\text{DR}\_{25}​$  | $\text{DR}\_{50}​$  |
> |----------|----------|----------|
> | SAR    | 0.307     | 0.201    |
> | TTR    | -0.028   | -0.005  |
> | Ours   | 0.044     | 0.031    |
>
> Under comparable parameter settings, we observed that SAR methods significantly exacerbate the degree of error accumulation. While both TTR and NAR effectively suppress error accumulation, NAR outperforms TTR in terms of generation speed and overall video quality. We hope these experiments will address your concerns regarding the comprehensiveness of our discussion on NAR.
>
> \
> \
> [4] Bian, Xiaohang, et al. "Handwritten mathematical expression recognition via attention aggregation based bi-directional mutual learning." Proceedings of the AAAI conference on artificial intelligence. Vol. 36. No. 1. 2022.

---

> ### Author Response · Authors · 2024-11-17
> **Author Response to Official Review by Reviewer fcWS (3/3)**
>
> > Q4：Methods like Anitalker, TH-PAD, and VASA-1, etc., have similar overall architectures with the proposed one, but the authors did not illustrate the differences between the proposed method and these methods.
>
>
> These works share commonalities with ours in that they all leverage diffusion models to generate motion representations, which are then reconstructed into image sequences through a decoder. Additionally, these methods also involve decoupling pose and lip movements. In contrast, our work differs from theirs in the following aspects:
>
> 1. **Non-Autoregressive Framework and Extrapolation Stability:**
>    We propose a fully non-autoregressive (NAR) framework for talking head generation, encompassing both NAR pose generation and NAR overall motion generation. Furthermore, we introduce a series of methods to enhance the extrapolation performance of NAR-based talking head video during inference, ultimately demonstrating the model's stable performance regardless of inference length. However, fully NAR generation and extrapolation stability are not within the scope of [6-8], nor did they apply specific optimizations to improve model extrapolation.
> 2. **Temporal Dependency Modeling:**
>    These works decouple lip and non-lip movements primarily to improve the quality and accuracy of facial motion generation. Additionally, we analyzed that the representation of pose movements is relatively simple but has longer durations and long-term dependency. Whereas the lip movement is more complex, with shorter durations and short-term dependency. Based on this analysis, we designed two sub-models to separately model the temporal dependencies of these two types of motion. This not only improves sequence generation quality but also ensures the stability of the model for long-sequence generation.
>
>
> > Q5：Recently, many diffusion-based talking head synthesis methods have been proposed, such as AniPortrait, Anitalker, Hallo, EchoMimic, FollowYourEmoji, and so on. Most of them operate in an auto-regressive manner. The authors did not compare the overall performance with these methods; therefore, there is no adequate evidence to prove the performance superiority of the proposed method.
>
>
>
> Thank you for your suggestion. Considering that these methods either use a large amount of data [8] or employ Stable Diffusion pre-trained models [2, 3, 5], we initially chose not to compare with them for the sake of rigor, as this could affect the fairness of the comparison. However, we agree that comparing with the latest DM-based models is indeed essential. Therefore, we have added more comparison experiments with the latest methods in the response to **Q1's comment**.
>
> > Q6：I have a question regarding the ablation study on the TCL strategy: when the model was trained only with stage 1 and only with stage 2, did you maintain the same number of training epochs and steps as in the proposed method? I am curious to know whether the effectiveness of the two-stage curriculum learning is due to the training time or the strategy itself.
>
> We adjusted the number of training epochs based on the amount of training data. When applying the TCL strategy, both stage 1 and stage 2 consist of 500 epochs. However, the average video length in stage 2 is nearly twice as long as in stage 1. Therefore, based on data throughput, in the ablation experiment, we trained for 1500 epochs (500 * 3) when only executing stage 1. When only executing stage 2, we trained for 800 epochs (slightly more than 500 * 1.5). We believe this setup will minimize the impact of different training epochs.
>
> \
> \
>
> [5] Ma, Yue et al. “Follow-Your-Emoji: Fine-Controllable and Expressive Freestyle Portrait Animation.” ArXiv abs/2406.01900 (2024): n. pag.
>
>
>
> [6] Yu, Zhentao, et al. "Talking head generation with probabilistic audio-to-visual diffusion priors." Proceedings of the IEEE/CVF International Conference on Computer Vision. 2023.
>
> [7] Xu, Sicheng, et al. "Vasa-1: Lifelike audio-driven talking faces generated in real time." arXiv preprint arXiv:2404.10667 (2024).
>
> [8] Liu, Tao, et al. "AniTalker: Animate Vivid and Diverse Talking Faces through Identity-Decoupled Facial Motion Encoding." arXiv preprint arXiv:2405.03121 (2024).

---

> ### Author Response · Authors · 2024-11-20
> **Thanks for your reply**
>
> Dear reviewer fcWS, thank you for taking the time and effort to read our rebuttal. We hope that our further responses can address your concerns.
>
>
> We didn't pre-generate the expression for the following two reasons:
>
> 1. **Decoupling Difficulty**
>
> Pose and blink are relatively independent of lip movements, exhibiting a lower degree of coupling. However, decoupling expressions from lip movements is significantly more challenging. This is because expressions and lip movements interact in numerous ways. For instance, speaking involves the movement of facial muscles, meaning that expressions need to integrate naturally with lip movements. Adopting a decoupling approach similar to pose and blink could compromise the natural dynamics of the face. Therefore, it is preferable for the model to generate lip movements and expressions simultaneously to ensure the naturalness and vividness of the results.
>
> 2. **Representation of Expression**
>
> Achieving high-fidelity representation of expressions is significantly more challenging compared to pose or blink. While some studies have attempted to represent expressions using 3D coefficients, such as in [1], these representations often compromise the naturalness of the generated expressions [2]. Therefore, we avoid adopting the same structure as PBNet for expression generation, as the representation of pose and blink is much simpler and does not lead to distortion issues.
> \
> \
> We acknowledge that the duration of expressions is generally longer compared to lip movements, and using a structure similar to PBNet for expression representation could potentially enhance the model's overall extrapolation ability. However, considering the challenges of achieving high-fidelity expression representation, the complexity of completely decoupling expressions, and the relatively shorter duration of expressions than pose/blink [1], we took these factors into account. To ensure the generation of natural, vivid videos with sufficient extrapolation capability, we opted to use a diffusion-based approach to simultaneously generate expressions and lip movements.
>
> [1] Zhang, Wenxuan et al. “SadTalker: Learning Realistic 3D Motion Coefficients for Stylized Audio-Driven Single Image Talking Face Animation.” 2023 IEEE/CVF Conference on Computer Vision and Pattern Recognition (CVPR) (2022): 8652-8661.
>
> [2] Tian, Linrui et al. “EMO: Emote Portrait Alive - Generating Expressive Portrait Videos with Audio2Video Diffusion Model under Weak Conditions.” ArXiv abs/2402.17485 (2024): n. pag.

---

> > ### Comment · Reviewer_fcWS · 2024-11-22
> > **Response to authors‘ rebuttal**
> >
> > I am aware of the difficulty of disentangling and the limitation of capability of off-the-shelf expression detectors. However, this is not a reason to omit expression modeling for NAR, and it contradicts with the authors' claim about modeling blinks and head pose using a NAR approach, as I described in my first response. Thus, I still believe that this paper does not meet the bar for publication because the proposed method does not properly support the authors' claim.
> > Considering that authors provide more extended experiments and design a new metric to quantify accumulation of errors, I still would like to raise my rating to 5.

---

> ### Author Response · Authors · 2024-11-22
> **We hope that our response addresses your concern**
>
> Dear Reviewer fcWS,
>
> We earnestly thank you for taking the time to read and respond to our rebuttal. We have carefully provided further explanation regarding your concern. We are eager to know if our response has addressed your concern. As the end of the rebuttal phase is approaching, we look forward to hearing from you for any further clarification that you might require.
>
> Best,
>
> Submission 6731 authors

---

> ### Author Response · Authors · 2024-11-23
> **Clarification of expression modeling**
>
> Dear Reviewer fcWS, thank you for acknowledging our experiments, and for providing us with many valuable and constructive suggestions that greatly helped us improve the quality of our paper. However, we respectfully maintain some reservations regarding your opinion that we "omit expression modeling for NAR, and it contradicts with the authors' claim about modeling blinks and head pose using a NAR approach."
>
> **We have not omitted expression modeling in NAR; instead, we use A2V-FDM for implicit modeling of expressions**. The core reason for this, as we mentioned earlier, is the difficulty of decoupling lips and expressions. Both PBNet and A2V-FDM operate in a NAR manner. Using A2V-FDM to generate expressions does not compromise the NAR property of the entire model. Moreover, in A2V-FDM, each frame is not generated independently and it possesses the sequence modeling capacity; we employ temporal attention interactions between frames (line 232) to effectively model the facial dynamics sequences, including expressions. Additionally, most current state-of-the-art AR methods also implicitly model expressions and lip motion [1,2,3]. The main reason is the same as we previously described. Decoupling expressions may have negative effects, so we believe the best approach at present is to avoid decoupling.
>
> Furthermore, our choice to decoupled modeling blink and pose is due to another reason: modeling pose/blink movements at the pixel level with a diffusion model needs training on sufficiently long video clips, which demands more resources than we currently have (line 059). To cut costs, we simplify the pose/blink sequences into low-dimensional forms and use lightweight PBNet for longer sequences training (200 frames). This ensures the modeling for pose and blink motion with long-term dependency. Unlike pose/blink, expressions lack significant long-term dependency, so we use clip up to 40 frames in training for joint motion modeling of lips and expressions by diffusion model in pixel-level. This is manageable with our computational resources. For tightly coupled expression and lip movements, pixel-level modeling is better for realistic facial dynamics generation. Our demos show our method can model lively expressions. Lastly, our design is proven to ensure stable extrapolation.
>
>
> [1] Xu, Sicheng et al. “VASA-1: Lifelike Audio-Driven Talking Faces Generated in Real Time.” ArXiv abs/2404.10667 (2024): n. pag.
>
> [2] Tian, Linrui et al. “EMO: Emote Portrait Alive - Generating Expressive Portrait Videos with Audio2Video Diffusion Model under Weak Conditions.” ArXiv abs/2402.17485 (2024): n. pag.
>
> [3] He, Tianyu et al. “GAIA: Zero-shot Talking Avatar Generation.” ArXiv abs/2311.15230 (2023): n. pag.

---

### Official Review · Reviewer_FboU · 2024-11-03

**Soundness:** 3
**Presentation:** 3
**Contribution:** 3
**Rating:** 8
**Confidence:** 4

**Summary:**

This work presents a non-autoreregrssive diffusion based approach for talking head generation. This allows the generation of videos with non-fixed length. To enhance the performance the authors also decouple the motion of lips, head, and blinks and also introduce a two stage curriculum learning strategy. The proposed model is evaluated on the CREMA and HTDF datasets achieving state-of-the-art results.

**Strengths:**

The paper is well written and easy to follow

First non-autoregressive diffusion-based approach for talking head generation

**Weaknesses:**

Why is the proposed approach compared against wav2lip? Wav2lip only generates the mouth region, therefore the comparison is not really fair. The authors are encouraged to explain in the paper this decision and clearly acknowledge this limitation.

The authors use the syncnet model for measuring the quality of the generated lip movements. Why not using a lipreading model? It would be a more accurate measure of the lip movement quality. An explanation why syncnet is preferred over lipreading models would be very useful. Alternatively, the authors could use some public lipreading models to evaluate the performance (e.g., AutoAVSR, Raven, AV-Hubert).

The authors use several quantitative metrics to evaluate the model's performance and this is good. However, a user study is missing. There are no quantitative metrics which correlate highly with human perception, therefore evaluating the performance of generative models via user studies is highly desirable. The paper would be much stronger if a user study is included. Otherwise, the authors should explain why it's not included.

Table 1, why are some results underlined? This should be explained in the captions.

On which dataset is the LFG trained? Is it trained on each dataset separately? Also, what's the impact of pre-training? Has the option of training the LFG component jointly with the rest of the model been considered (and if yes can some results be presented)? The paper would be stronger if the author provide additional details regarding the above questions.

**Questions:**

Please see above.

---

> ### Author Response · Authors · 2024-11-17
> **Author Response to Official Review by Reviewer FboU (1/3)**
>
> Dear reviewer FboU, we are grateful for your careful review and precious advices to our work. We hope our following comments address your concerns. Additionally, we have uploaded new supplementary videos to the [anonymous repository](https://anonymous.4open.science/r/DAWN-BC9F/README.md). Since the files are large, for a better viewing experience, we recommend downloading the supplementary materials first.
>
> > Q1 : Why is the proposed approach compared against wav2lip? Wav2lip only generates the mouth region, therefore the comparison is not really fair.&#x20;
>
> Although the wavlip method only generates mouth movements, it performs as the best model in LSE-C and LSE-D metrics. Therefore, we aim to use wav2lip as a reference on these two metrics to better assess DAWN's performance in generating lip movements. To avoid misunderstanding, we have clarified it in the title of Table 1 and marked it with an asterisk (*). We will explain wavlip’s detailed settings more explicitly in the comparative experiments section.
>
> > Q2: The authors use the syncnet model for measuring the quality of the generated lip movements. Why not using a lipreading model? It would be a more accurate measure of the lip movement quality.&#x20;
>
> Thank you for your suggestion. The reason we use the SyncNet-based LSE score is that it is a mainstream metric for evaluating lip movement in talking head generation and plays a key role in assessing the quality of generated results [1,2,3,4,5,6,7]. While existing works have less frequently used utilized lip-reading models for evaluating lip movement quality, we agree that lip-reading models have potential in this regard. Their extensive pretraining can enable a more accurate understanding of lip movements. Additionally, directly using a lipreading model may present some difficulties because SyncNet is used to determine the consistency between audio and lip movements, while a lipreading model performs direct lip reading, a task much more complex than SyncNet's. Therefore, it appears that lipreading models may currently lack the robustness required to evaluate arbitrary videos, particularly in providing accurate assessments for synthesized lips that contain errors. Therefore, we plan to explore the feasibility of using lip-reading models for evaluation in future work.
>
> [1] Stypułkowski, Michał, et al. "Diffused heads: Diffusion models beat gans on talking-face generation." Proceedings of the IEEE/CVF Winter Conference on Applications of Computer Vision. 2024.
>
> [2] Zhang, Wenxuan, et al. "Sadtalker: Learning realistic 3d motion coefficients for stylized audio-driven single image talking face animation." Proceedings of the IEEE/CVF Conference on Computer Vision and Pattern Recognition. 2023.
>
> [3] Tian, Linrui, et al. "Emo: Emote portrait alive-generating expressive portrait videos with audio2video diffusion model under weak conditions." arXiv preprint arXiv:2402.17485 (2024).
>
> [4] Xu, Sicheng, et al. "Vasa-1: Lifelike audio-driven talking faces generated in real time." arXiv preprint arXiv:2404.10667 (2024).
>
> [5] Xu, Mingwang, et al. "Hallo: Hierarchical Audio-Driven Visual Synthesis for Portrait Image Animation." arXiv preprint arXiv:2406.08801 (2024).
>
> [6] Shen, Shuai, et al. "Difftalk: Crafting diffusion models for generalized audio-driven portraits animation." Proceedings of the IEEE/CVF Conference on Computer Vision and Pattern Recognition. 2023.
>
> [7] Ma, Yifeng, et al. "Dreamtalk: When expressive talking head generation meets diffusion probabilistic models." arXiv preprint arXiv:2312.09767 (2023).

---

> > ### Comment · Reviewer_FboU · 2024-11-26
> >
> > I would like to thank the authors for their reply.
> >
> > Q2: It is true that past works have used syncnet, but one of the main reasons for this is the lack of available lipreading models in the past. At the moment, there are several SOTA lipreading models which are publicly available.

---

> > > ### Author Response · Authors · 2024-11-27
> > > **Thanks for your reply**
> > >
> > > Dear Reviewer FboU,
> > >
> > > We sincerely appreciate your response and valuable suggestion regarding the use of the lipreading model for access. We will commit to actively exploring its application in evaluating model performance. Furthermore, inspired by your insightful advice, we are keen to investigate the possibility of assessing lip movement quality by analyzing the similarity of latent features extracted by the lipreading model, similar to the concept of FID. We are actively trying to include lipreading metric in the final PDF version. As the deadline for updating the paper PDF is approaching, we have already added our explanation of Q5 in the latest version in the chapter of Training Details (Appendix A.2, line 775) and Limitations and Future Works (Appendix A.4, line 962). Finally, we extend our heartfelt gratitude again for your support in enhancing the quality of our paper.
> > >
> > > Best,
> > >
> > > Submission 6731 authors

---

> ### Author Response · Authors · 2024-11-17
> **Author Response to Official Review by Reviewer FboU (2/3)**
>
> > Q3： The paper would be much stronger if a user study is included. Otherwise, the authors should explain why it's not included.
>
> Thank you for your suggestion. We recognize that human evaluation is indeed indispensable, so we conducted the **user study**. The study evaluates the generated videos from four dimensions: 1) **L-Sync**: Lip-audio synchronization; 2) **O-Nat**: The overall naturalness of the generated talking head ; 3) **M-Viv**: The vividness of the head movements; 4) **V-Qual**: The overall video quality (e.g., presence of artifacts or abnormal color blocks). We generated 10 test videos for each method, and a total of 23 participants took part in the user study, scoring each dimension on a scale of 1 to 5. To strengthen the comparison, we added several recent diffusion-based  talking head methods to the original baselines, including **DreamTalk (2023.12)** [7], **Hallo (2024.6)** [5], and **EchoMimic (2024.7)** [8]. Since these methods vary in resolution and some employ image cropping to centralize the face [7,8], we instructed users to disregard resolution and cropping during testing when providing their ratings.
>
>
> As shown in the results, our method outperforms existing approaches in lip synchronization, overall naturalness, and head movement vividness. In terms of video quality, our approach is comparable to the concurrent work EchoMimic [8]. We will include the user study as a section in the experiment part of the paper.
>
> |           | L-Sync | O-Nat | M-Viv |V-Qual |
> |----------|----------|----------|----------|----------|
> | Audio2Head| 3.87     | 3.67     |3.28      |3.66      |
> | Sadtalker| 4.23     | 3.13     |2.81      |4.14      |
> | DreamTalk   | 4.38     | 3.89     |3.41      |4.42      |
> | Hallo   | 4.40     | 3.76     |3.89      |4.03      |
> | Echomimic   | 4.45     | 4.30     |4.06      |**4.53**      |
> | DAWN(ours)   | **4.57**     | **4.41**     |**4.43**      |4.51      |
>
> Additionally, we have supplemented our work with a **quantitative comparison** against the newly added methods. The results demonstrate that our method achieves competitive performance compared to the state-of-the-art, both in subjective and objective metrics. It is worth noting that although Hallo achieved high FID/FVD scores in the quantitative study, it received lower scores in human evaluations. This is because humans are particularly sensitive to quality degradation resulting from error accumulation, which has been visualized in our supplement material "[Compare_with_Hallo.mp4](https://anonymous.4open.science/r/DAWN-BC9F/Compare_with_Hallo.mp4)".
>
> |          | FID      | FVD16|FVD32|LSE-C| LSE-D| CSIM     | BAS      | Blink/s  |
> |----------|----------|----------|----------|----------|----------|----------|----------|----------|
> | GT       | -        | -        | -        | 7.95     | 7.33     | 1       | 0.267     | 0.75     |
> | DreamTalk| 58.8     | 406.58     | 516.21     | 6.48     | 8.43     | 0.641     | 0.3110     | 0.032     |
> | Hallo     | 14.2     | **57.47**     | 100.99     | **7.16**     | 8.01     | 0.7095     | 0.3010     | 0.254     |
> | Echomimic| 32.8     | 139.00     | 178.16     | 6.69     | 8.27     | 0.7314     | 0.3177     | 0.1212     |
> | DAWN(ours)    | **9.60**    | 60.34     | **95.64**    | 6.71     | **7.94**    | **0.790**     | **0.281**     | **0.86**     |
>
> > Q4：Table 1, why are some results underlined? This should be explained in the captions.
>
> Thank you for your careful review. The underlined text is intended to indicate the second-best results. We will add an explanation in the caption of Table 1 to avoid potential confusion.
>
> \
> \
> [8] Chen, Zhiyuan, et al. "Echomimic: Lifelike audio-driven portrait animations through editable landmark conditions." arXiv preprint arXiv:2407.08136 (2024).

---

> ### Author Response · Authors · 2024-11-17
> **Author Response to Official Review by Reviewer FboU (3/3)**
>
> > Q5: On which dataset is the LFG trained? Is it trained on each dataset separately? Also, what's the impact of pre-training? Has the option of training the LFG component jointly with the rest of the model been considered (and if yes can some results be presented)?
>
> The LFG is fine-tuned independently on the training datasets of CREMA and HDTF, using a checkpoint initially trained on the VoxCeleb dataset. During the training of DAWN, LFG provides supervision information for A2V-FDM. The purpose of pre-training LFG is to supply A2V-FDM with the most precise supervision signals possible, ensuring its convergence.
>
> Thank you for your suggestion. The primary reason we did not adopt an end-to-end training approach is the difficulty in achieving stable convergence. In an end-to-end training setup, the DM is likely to receive incorrect supervision signals, which can hinder the training process of the diffusion model and make it highly unstable. In the end-to-end training experiment, we found that despite using some training techniques such as gradient clipping and using a lower learning rate, the loss function typically diverged within 1000 iterations during the training process. Therefore, we believe that there are certain obstacles to end-to-end training under the current model structure. However, end-to-end model structures might prevent the accumulation of errors across multiple stages, which we believe is indeed worth researching. Therefore, we will explore the approach to enhance the stability of joint training in the future.

---

> ### Author Response · Authors · 2024-11-22
> **We hope that our response addresses your concern**
>
> Dear Reviewer FboU,
>
> We greatly appreciate the time you've invested in reviewing our response. Having submitted our rebuttal, we are eager to know if our response has addressed your concern. As the end of the rebuttal phase is approaching, we look forward to hearing from you for any further clarification that you might require.
>
> Best,
>
> Submission 6731 authors

---

> ### Comment · Reviewer_FboU · 2024-11-26
>
> Please add the above explanation (for Q5) in the paper.

---

### Author Response · Authors · 2024-11-19
**General Comments to All Reviewers**

We sincerely thank all reviewers for their valuable suggestions and efforts. We have earnestly responded to each suggestion and accordingly revised our paper. If you find our answers responsive to your concerns, we would be grateful if you would consider increasing your score.

In response to the reviewers' suggestions and comments, we have revised the manuscript, with all changes marked in blue. These revisions include:
- Main Paper: In Equation 6 (line 226), we correct some of the imprecise expressions. (Reviewer  Q1nh )
- Main Paper: In Section 3.6 (line 311), emphasize both the PBNet and A2V-FDM generate the sequence with dynamic length non-autoregressively, and the generation length depends on input audio length. (Reviewer Q1nh)
- Main Paper: In Table 1 (line 381), we add the details for the comparison with Wav2Lip, and explain why Wav2Lip didn't participate in ranking. (FboU, Q1nh) Also, we explain the meaning of \underline{underline} (line 380). (Reviewer  FboU)
- Main Paper: In Table 1 (line 398) and Figure 2 (line 449), we add **three** latest diffusion-based methods to reinforce the experiment in the overall comparison section. (Reviewer  fcWS, Q1nh)
- Main Paper: In Section 4.4 (line 349), Table 2 (line 409) and Table 6 (line 834), we add the **Degradation  Rate** for evaluation of error accumulation, to better prove the superiority of our non-autoregressive method. At the same time, we have added multiple citations (line 46) to support the assertion that AR methods lead to error accumulation. (Reviewer  fcWS, Q1nh, 5zi1)
- Main Paper: In Section 4.5 (line 499), we add the **user study** to reinforce our experiment by human evaluation. (Reviewer  FboU, 5zi1)
- Appendix: In Appendix A.2 (line 768), we add the Training Details to better explain the configuration in training (including LFG, PBNet and A2V-FDM). (Reviewer FboU, fcWS, Q1nh)
- Appendix: In Appendix A.3.3 (line 858), we add the quantitative study on our method with higher resolution. (Reviewer Q1nh)
- Appendix: In Appendix A.3.4 (line 933), we add two latest diffusion-based methods for image latent into generation speed comparison. (Reviewer 5zi1)

- Main Paper: We specify that the FID-based Degradation Rate proposed in the paper is specifically designed for detecting the degradation problem of image-based AR diffusion methods, as it focuses on the degradation detection in the image space. (Main Paper line 349, Appendix line 801, 807) (Reviewer Q1nh)

Please kindly check our updated manuscript. We welcome further discussion.

---

### Meta-Review · Area_Chair_NfHY · 2024-12-22

**Metareview:**

This paper received mixed scores: one borderline accept, two borderline reject, and one accept. Most reviewers acknowledged the simple and effective solution proposed in the paper. Specifically, the paper introduces the first non-autoregressive diffusion-based approach for talking head generation, which was recognized as a novel contribution by the reviewers. However, some reviewers raised concerns, including the high similarity to existing methods such as Anitalker, TH-PAD, and VASA-1, limited original contributions, and restricted practicality of the approach. The AC has reviewed the paper, the reviews, and the rebuttal. While there have been numerous papers on generative model-based talking face synthesis, such as VASA-1, the application of non-autoregressive diffusion (NAD) to this task is novel and was acknowledged by the reviewers. Furthermore, the paper is well-written and presents its ideas clearly. Therefore, the AC suggests acceptance. It would be even better if the authors addressed all the reviewers' concerns in the final version.

**Additional Comments On Reviewer Discussion:**

There were no discussion.

---

### Decision · Program_Chairs · 2025-01-22

Accept (Poster)